# The intrinsically disordered cytoplasmic tail of a dendrite branching receptor uses two distinct mechanisms to regulate the actin cytoskeleton

Daniel A Kramer[1], Heidy Y Narvaez-Ortiz[2], Urval Patel[1], Rebecca Shi[3,4], Kang Shen[3,5], Brad J Nolen[2], Julien Roche[1]*, Baoyu Chen[1]*

[1]Roy J Carver Department of Biochemistry, Biophysics, and Molecular Biology, Iowa State University, Ames, United States; [2]Department of Chemistry and Biochemistry, Institute of Molecular Biology, University of Oregon, Eugene, United States; [3]Department of Biology, Stanford University, Stanford, United States; [4]Neurosciences IDP, Stanford University, Stanford, United States; [5]Howard Hughes Medical Institute, Stanford University, Stanford, United States

*For correspondence:
jroche@iastate.edu (JR);
stone@iastate.edu (BC)

**Abstract** Dendrite morphogenesis is essential for neural circuit formation, yet the molecular mechanisms underlying complex dendrite branching remain elusive. Previous studies on the highly branched *Caenorhabditis elegans* PVD sensory neuron identified a membrane co-receptor complex that links extracellular signals to intracellular actin remodeling machinery, promoting high-order dendrite branching. In this complex, the claudin-like transmembrane protein HPO-30 recruits the WAVE regulatory complex (WRC) to dendrite branching sites, stimulating the Arp2/3 complex to polymerize actin. We report here our biochemical and structural analysis of this interaction, revealing that the intracellular domain (ICD) of HPO-30 is intrinsically disordered and employs two distinct mechanisms to regulate the actin cytoskeleton. First, HPO-30 ICD binding to the WRC requires dimerization and involves the entire ICD sequence, rather than a short linear peptide motif. This interaction enhances WRC activation by the GTPase Rac1. Second, HPO-30 ICD directly binds to the sides and barbed end of actin filaments. Binding to the barbed end requires ICD dimerization and inhibits both actin polymerization and depolymerization, resembling the actin capping protein CapZ. These dual functions provide an intriguing model of how membrane proteins can integrate distinct mechanisms to fine-tune local actin dynamics.

## Editor's evaluation

This important work investigates a dual mechanism of regulation of actin assembly by the dendrite branching receptor HPO-30. Convincing data demonstrate how oligomerization of HPO-30 ICD impacts the activity of the Wave Regulatory Complex (WRC), but also that direct binding of HPO-30 to filamentous actin modulates actin polymerization properties. This work sheds light on a crucial biological question but also shows the importance of the issue of conformation and oligomerization of disordered proteins in the regulation of actin regulatory mechanisms.

## Introduction

Neurons exhibit a variety of shapes, but they all share a fundamental architecture comprising a cell body, a primary axon, and many branching dendrites adorned with numerous tiny projections called

spines (*Burianek and Soderling, 2013*; *Jan and Jan, 2003*; *Lefebvre et al., 2015*). Dendrite and spine formation is critical for nervous system development, as they establish the majority of post-synaptic connections in animals and define the intricate wiring of neural circuits (*Hotulainen and Hoogenraad, 2010*; *Scott and Luo, 2001*; *Tavosanis, 2012*). Actin and microtubule filaments are essential for initiating and promoting the outgrowth of newly formed neurites during dendrite development (*Cheng and Poo, 2012*; *Zhao et al., 2017*). Disruptions in actin cytoskeleton regulation can alter dendrite morphology and neural connections, contributing to various neurodevelopmental disorders such as autism, mental retardation, and schizophrenia (*Yan et al., 2016*).

A primary mechanism by which neurons regulate actin polymerization involves the Arp2/3 complex, a seven-protein assembly that binds to the sides of existing actin filaments and promotes the formation of branched actin filament networks (*Goley and Welch, 2006*; *Hasegawa et al., 2022*; *Machesky et al., 1994*; *Pollard, 2007*). The Arp2/3 complex is intrinsically inactive and requires activation by nucleation promoting factors, which primarily consist of the Wiskott-Aldrich syndrome protein (WASP) family proteins (*Alekhina et al., 2017*; *Kramer et al., 2022*; *Pollitt and Insall, 2009*; *Takenawa and Suetsugu, 2007*). These proteins contain a conserved WCA (WH2-central-acidic) sequence at their C terminus, which can directly bind to and activate Arp2/3. The N-terminal sequences of these proteins vary greatly and define their regulatory mechanism and cellular function (*Alekhina et al., 2017*; *Kramer et al., 2022*; *Machesky and Insall, 1998*).

WAVE (Wiskott-Aldrich verprolin homology) belongs to the WASP family and is part of a multi-protein assembly known as the WAVE regulatory complex (WRC). Consisting of five subunits—Sra1, Nap1, Abi2, HSPC300, and WAVE (or their corresponding orthologs in vertebrates) (*Chen et al., 2010*; *Eden et al., 2002*; *Gautreau et al., 2004*; *Polesskaya et al., 2021*; *Rottner et al., 2021*)—the WRC facilitates actin polymerization downstream of membrane signaling in neurons. This activity is crucial for various neuronal processes, including growth cone formation, axon branching, dendrite branching, synapse formation, and axon guidance and projection (*Chia et al., 2014*; *Chou and Wang, 2016*; *Pilpel and Segal, 2005*; *Rácz and Weinberg, 2008*; *Soderling et al., 2007*; *Stephan et al., 2011*; *Tahirovic et al., 2010*; *Yamazaki et al., 2005*; *Yokoyama et al., 2011*). Disruption of WRC function can profoundly affect the nervous system in animals, resulting in altered spine morphology and density, intellectual disability, and embryonic death (*Dahl et al., 2003*; *Soderling et al., 2003*). Mutations in WAVE and other WRC subunits have been linked to various neurodevelopmental disorders in humans, including neurodevelopmental disorder with absent language and variable seizures, developmental and epileptic encephalopathy-65, and Alzheimer's disease (*Begemann et al., 2021*; *Conway et al., 2018*; *Ito et al., 2018*; *Kirkpatrick et al., 2017*; *Kramer et al., 2022*; *Kumar et al., 2013*; *Olive et al., 2020*; *Rottner et al., 2021*; *Shimojima Yamamoto et al., 2021*; *Sims et al., 2017*; *Srivastava et al., 2021*; *Zhao et al., 2021*; *Zweier et al., 2019*).

The WRC inhibits WAVE activity by sequestering its C-terminal WCA to a conserved surface formed by the Sra1 and WAVE subunits (*Chen et al., 2017*; *Chen et al., 2010*; *Eden et al., 2002*; *Kramer et al., 2022*). Various molecules can recruit the WRC to the membrane and/or simultaneously activate it through direct interactions, releasing the WCA to promote Arp2/3-mediated actin polymerization. These ligands include small GTPases (e.g. Rac1 and Arf), acidic phospholipids (e.g. PIP$_3$), adaptor proteins, and a wide array of membrane proteins containing the WIRS motif (WRC interacting receptor sequence, defined as $\Phi$-x-T/S-F-x-x, where $\Phi$ is a bulky hydrophobic residue and x is any residue) (*Chen et al., 2017*; *Chen et al., 2010*; *Ding et al., 2022*; *Eden et al., 2002*; *Kobayashi et al., 1998*; *Koronakis et al., 2011*; *Lebensohn and Kirschner, 2009*; *Rottner et al., 2021*; *Yang et al., 2022*). Many WIRS-containing membrane proteins, such as SYG-1, Robo, DCC, Neogenin, TMEM132, neuroligins, and various protocadherins, are essential neuronal receptors that depend on the WIRS-WRC interaction to regulate diverse neural development processes (*Chaudhari et al., 2022*; *Chaudhari et al., 2021*; *Chia et al., 2014*; *Fan et al., 2018*; *Lee et al., 2016*; *Wang et al., 2021*; *Xing et al., 2018*).

In previous studies, the claudin-like receptor HPO-30 was identified as a novel WRC-binding protein, which does not contain a WIRS motif. The HPO-30-WRC interaction is essential for high-order dendrite branching in the PVD sensory neuron in *Caenorhabditis elegans* (*Zou et al., 2016*). HPO-30 acts as a co-receptor with the cell adhesion molecule DMA-1. The extracellular domain of DMA-1 forms a multi-ligand complex with the secreted protein LECT-2 and the extracellular domains of the epidermal cell receptors SAX-7 and MNR-1, providing spatial cues for initiating dendrite branching in

PVD neurons (*Dong et al., 2013*; *Zou et al., 2018*; *Zou et al., 2016*). The intracellular domain (ICD) of HPO-30 directly binds to the WRC, while the ICD of DMA-1 binds the Rac guanine nucleotide exchange factor (GEF) TIAM-1, increasing the local concentration of activated Rac1, the canonical activator of WRC (*Tang et al., 2019*; *Zou et al., 2018*). Together, the HPO-30-DMA-1 co-receptor organizes a multi-protein complex that bridges extracellular cues to the intracellular actin remodeling machinery to regulate high-order dendrite branching (*Zou et al., 2018*).

The interaction mechanism between HPO-30 ICD and the WRC remains elusive, and it is uncertain if the ICD has additional functions beyond its WRC binding activity. Unlike other WRC-binding membrane proteins, HPO-30 ICD lacks a WIRS motif and uses a non-WIRS mechanism for WRC interaction. Prior research has shown that deletion of the C-terminal half of the HPO-30 ICD strongly affects WRC binding in vitro and high-order dendrite branching in vivo. However, it is unclear whether the HPO-30 ICD uses a linear peptide motif analogous to the WIRS motif for WRC binding (*Zou et al., 2018*). Although HPO-30 shares homology with mammalian tight junction claudin proteins, its ICD sequence is conserved only in nematode worms. Intriguingly, HPO-30 ICD binds to both *C. elegans* and human WRC, suggesting a conserved interaction mechanism across species (*Zou et al., 2018*). Thus, understanding the HPO-30-WRC interaction is critical for identifying other potential membrane proteins that regulate the WRC in humans through the same interaction surface.

In this study, we conducted biochemical and structural analyses of the HPO-30-WRC interaction. Our findings reveal that the HPO-30 ICD differs from other WRC-interacting receptors, as the entire ICD sequence—rather than a short peptide motif like the WIRS—is involved in the interaction. We also discovered that dimerization is necessary for HPO-30 ICD to bind to the WRC, which works cooperatively with Rac1 to promote WRC activation. Surprisingly, we found that the HPO-30 ICD directly interacts with actin filaments in a dimerization-dependent manner. The dimeric form of HPO-30 ICD binds to both the side and the barbed end of actin filaments, inhibiting both actin polymerization and depolymerization, resembling the activity of the actin capping protein CapZ. These dual activities of HPO-30 ICD present a fascinating example of how a membrane receptor can regulate local actin dynamics by simultaneously controlling the localization and activity of a central actin nucleation factor and directly binding to and modulating actin filaments to support an essential biological process.

## Results

### HPO-30 ICD uses the entire sequence, rather than a short peptide motif, to bind the WRC

The HPO-30 ICD lacks a WIRS motif, which led us to hypothesize that it might use a different linear motif to bind the WRC. To identify this potential sequence motif, we divided HPO-30 ICD sequence into continuous segments of five amino acids (a.a.) (Δ1–10 in *Figure 1A*) and mutated all a.a. in each segment to alanines. We then used GST pull-down assays to assess the impact of each segment on WRC binding (*Figure 1B*). Interestingly, we found that mutating nearly any segment reduced the binding to WRC, with the middle region of the ICD showing the most significant effect (Δ5, -6, -7 in *Figure 1A and B*, lanes 7–9). This finding supports the previous qualitative data obtained under suboptimal pull-down conditions (*Zou et al., 2018*). It is worth noting that while most mutants contained an N-terminal GST-tag, several mutations near the C-terminus of the ICD used a C-terminal GST-tag to improve the yield of full-length protein (Δ8–10 in *Figure 1*, lanes 10–13 in *Figure 1B*).

It is interesting that none of the alanine scan mutations completely abolished the binding to WRC, unlike WIRS-mediated binding where single a.a. mutations in the WIRS motif can readily eliminate the interaction (*Chen et al., 2014b*). In line with this finding, the wild type (WT) ICD had a dissociation constant ($K_D$) of ~1.69 μM, as measured by an equilibrium pull-down (EPD) assay (*Figure 1D*, black, and *Figure 1—figure supplement 1A*), while the alanine scan mutant Δ5, which had the strongest effect in the non-equilibrium GST pull-down assay (*Figure 1A and B*, lane 7), only mildly increased the $K_D$ to ~5.06 μM, still maintaining significant residual binding (*Figure 1D*, light yellow, and *Figure 1—figure supplement 1B*; *Chen et al., 2017*; *Kuzmic, 1996*; *Pollard and Kellogg, 2010*). More extensive mutations in the ICD, either in the N-terminal, middle, or C-terminal regions, more severely disrupted the binding (*Figure 1A and C* Δ11–14). These data suggest that the entire ICD sequence is involved in interacting with the WRC. This is distinct from WIRS-containing membrane proteins where

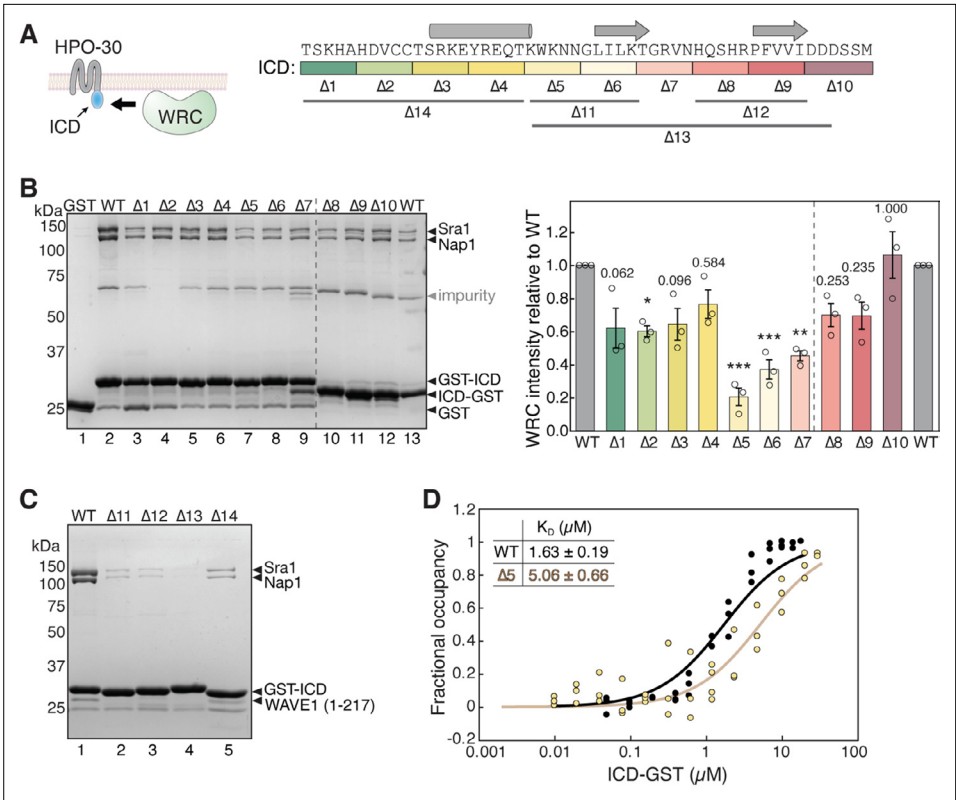

**Figure 1.** HPO-30 intracellular domain (ICD) binds to the WAVE regulatory complex (WRC) using its entire sequence. (**A**) Left: Cartoon representation of HPO-30 as a four-pass transmembrane protein, with the ICD binding to the WRC. Right: Annotation of HPO-30 ICD sequence and mutants used in this study. In each mutant, the corresponding amino acids were replaced with alanines (for Δ1–10) or (GGS)n (for Δ11–14). The regions are color-coded according to the alanine mutations, and the consensus secondary structural elements from multiple secondary structure predictions are indicated above. (**B**) Representative Coomassie blue-stained SDS-PAGE gel (left) and quantification (right) of three independent experiments showing GST-ICD (lanes 2–9, 200 pmol) and ICD-GST (lanes 10–13, 200 pmol) pulling down WRC$^{230\Delta WCA}$ (150 pmol). Sra1/Nap1 band intensity was used to quantify the pull-down signals of WRC. Signals from GST-ICD or ICD-GST pull-downs were normalized to the corresponding wild-type (WT) ICD (lanes 2 and 13, respectively). Error bars represent standard error, *p<0.05, **p<0.005, ***p<0.001 from ANOVA with Tukey test. n = 3. (**C**) Coomassie blue-stained SDS-PAGE gel showing GST-ICD Δ11–14 mutants (200 pmol) pulling down WRC$^{217\Delta WCA}$ (300 pmol). (**D**) Fitted curves of the dissociation constant ($K_D$) of the interaction between ICD-GST and WRC. Data were pooled from three independent repeats and globally fitted to a one-site binding model using DynaFit (n = 3). The raw gel images and unedited figure can be found in *Figure 1—source data 1*. The raw data for (**B**, right, and **D**) can be found in *Figure 1—source data 2*.

The online version of this article includes the following source data and figure supplement(s) for figure 1:

**Source data 1.** This folder contains the unedited *Figure 1* and raw gel images used in *Figure 1*.

**Source data 2.** This folder contains the raw data and statistical analysis shown in *Figure 1B and D*.

**Figure supplement 1.** Equilibrium pull-down (EPD) assay of intracellular domain (ICD)-GST and WAVE regulatory complex (WRC).

**Figure supplement 1—source data 1.** This folder contains the unedited *Figure 1—figure supplement 1* and raw gel images used in *Figure 1—figure supplement 1*.

**Figure supplement 2.** Ab initio structure prediction of HPO-30 intracellular domain (ICD).

**Figure supplement 2—source data 1.** This folder contains the unedited *Figure 1—figure supplement 2* and a Pymol scene file containing all predicted structural models shown in *Figure 1—figure supplement 2* aligned to one another.

**Figure supplement 3.** Molecular dynamics (MD) simulations of predicted intracellular domain (ICD) structures.

**Figure supplement 3—source data 1.** This folder contains the unedited *Figure 1—figure supplement 3* and separate folders containing individual Pymol scenes of the starting and final conformations for all models used in molecular dynamics (MD) simulations and root mean square deviation (RMSD) values for each simulation.

the WIRS motif is essential for binding the WRC, and sequences outside the WIRS motif have only marginal contributions, if any (*Chen et al., 2014b*).

## HPO-30 ICD contains residual structural elements, but is predominantly disordered in solution

Two potential models could explain how the entire ICD of HPO-30 binds to the WRC: either the ICD might fold into a three-dimensional structure to bind, or the ICD might exist as an extended linear peptide that makes an extensive contact with the WRC. To distinguish between these two models and understand how the binding occurs, we investigated the structure of the ICD using various approaches.

First, we submitted the sequence of ICD to multiple in silico structural prediction software. These programs include secondary structure analyses (JPred, PredictProtein, PSIPred, and SABLE) and ab initio tertiary structure predictions (LOMETS, QUACK, RaptorX, Rosetta, AlphaFold 2.0, and RoseT-TAFold) (*Figure 1—figure supplement 2*; *Adamczak et al., 2004*; *Bernhofer et al., 2021*; *Bonneau et al., 2001*; *Drozdetskiy et al., 2015*; *Jumper et al., 2021*; *Källberg et al., 2012*; *McGuffin et al., 2000*; *Minkyung et al., 2021*; *Wu and Zhang, 2007*). Nearly all programs predicted that the ICD contains a short alpha helix followed by two short beta strands (*Figure 1A*). This is distinct from WIRS-containing proteins, where the WIRS motif must reside in unstructured sequences for proper binding. It is noteworthy that the sequences most critical for binding, such as Δ5, Δ6, and Δ7 in *Figure 1*, are within or close to these secondary structural elements. In contrast to the consistent secondary structure prediction, however, tertiary structure predictions yielded varied results, in which the secondary structural elements are folded differently depending on the program and the context of ICD sequence (isolated ICD vs. full-length HPO-30) used for the predictions (*Figure 1—figure supplement 2*). The variability observed in the predictions suggests that the ICD may not adopt a well-defined three-dimensional structure, but rather may exist as a large and heterogeneous conformational ensemble.

To further assess these structural predictions, we performed a series of all-atom molecular dynamics (MD) simulations using the AMBER99SB-disp force field, which combines a new generation of water model with parameters specifically designed to capture the behavior of partially disordered proteins in solution (*Robustelli et al., 2018*). The top-ranking conformations from all eight predictions were subjected to 200 ns simulations to assess the conformational stability of the predicted structural models. The trajectories showed significant conformational fluctuations, with large root mean square deviation (RMSD) up to 20 Å from their corresponding initial models (*Figure 1—figure supplement 3*). The snapshots of the simulated molecules revealed major random reorganization of the structural elements, suggesting that the predicted tertiary structures were transient or unstable conformations, or may require specific conditions or contexts to exist. Notably, in almost all simulated structures, the alpha-helical region remained relatively stable, supporting its presence in the largely disordered ICD (*Figure 1—figure supplement 3*). Taken together, these in silico analyses suggest that while the ICD may contain structural elements, it is likely disordered and does not possess a stable conformation in solution.

To experimentally validate the above computational analyses, we used circular dichroism (CD) spectroscopy to evaluate HPO-30 ICD secondary structure composition in solution. In agreement with the MD simulations, the CD spectrum of ICD did not display major peaks at the wavelengths characteristic of either alpha helices (positive at 193 nm and negative at 218 and 222 nm) or beta sheets (positive at 195 nm and negative at 217 nm), clearly distinct from the CD spectrum of bovine serum albumin (BSA) obtained under the same conditions (*Figure 2A*). Adding an osmolyte, trimethylamine N-oxide (TMAO), which is commonly used to promote protein folding, did not significantly alter the CD spectrum or enhance the ICD-WRC interaction (*Figure 2—figure supplement 1A and B*; *Baskakov et al., 1999*). Yet, careful examination of the CD spectrum revealed that the ICD is different from unstructured proteins, which typically exhibit a positive peak at 222 nm (*Hwang and Waite, 2012*). This suggests that the HPO-30 ICD might contain some structural elements, possibly a helical region, as reflected by the negative peak at 222 nm.

Being unable to grow crystals of the ICD (either by itself, attached to solubility tags, or together with the WRC), we used solution nuclear magnetic resonance (NMR) spectroscopy to further investigate the structural features of the ICD. The $^1$H-$^{15}$N heteronuclear single quantum coherence (HSQC) spectra of untagged $^{15}$N-labeled ICD were well resolved, from which we identified 50 distinct amide crosspeaks, out of the 51 expected (*Figure 2B*). The narrow $^1$H chemical shift dispersion was consistent with our

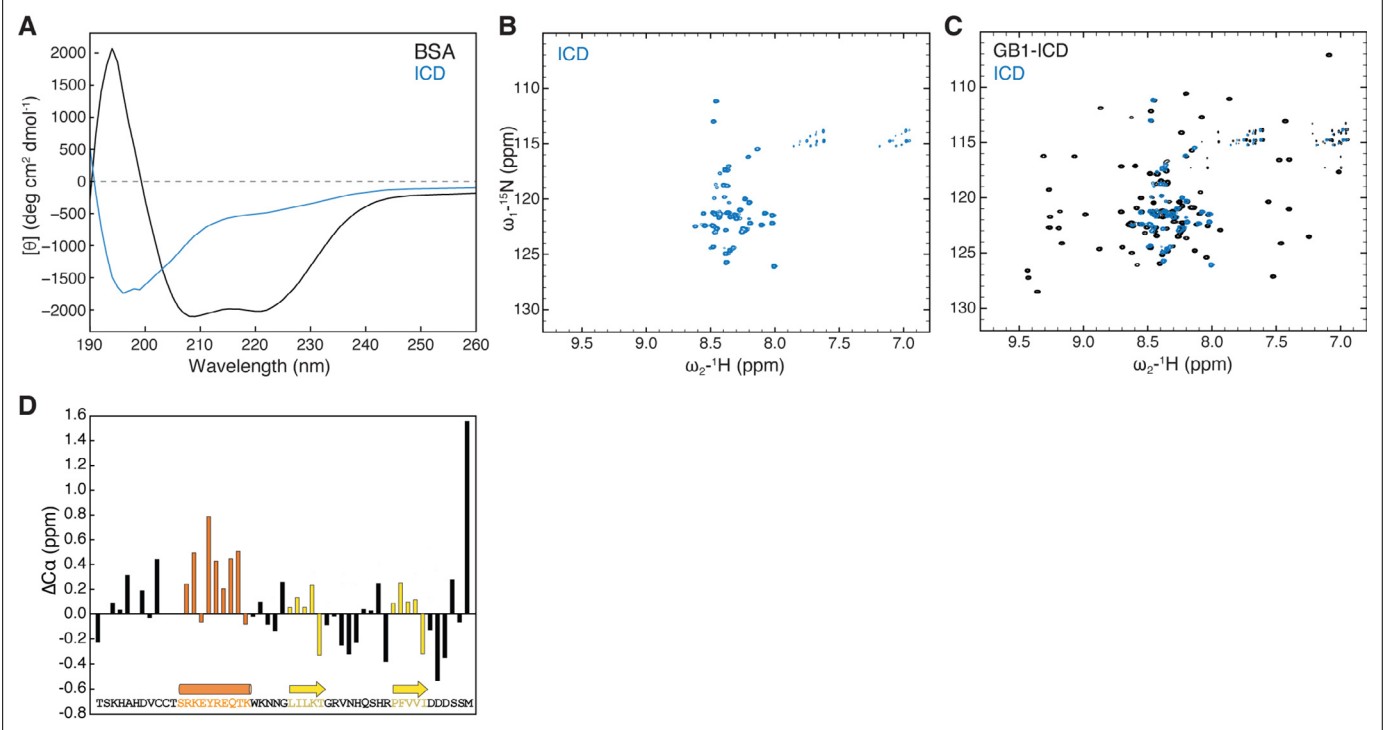

**Figure 2.** HPO-30 intracellular domain (ICD) is largely disordered but may contain secondary structural elements. (**A**) CD spectrum of 2 mg/mL untagged HPO-30 ICD or bovine serum albumin (BSA) in identical buffer conditions (see Materials and methods). (**B–C**) 2D TROSY spectra of untagged HPO-30 ICD (**B**) and GB1-ICD (**C**) in the same buffer condition (100 mM NaCl, 10 mM HEPES pH 7.0, and 5% glycerol). (**D**) Secondary Cα chemical shift showing the deviation from idealized random coils of Cα chemical shifts measured for HPO-30 ICD. The raw CD data can be found in *Figure 2— source data 1*, the raw nuclear magnetic resonance (NMR) data in (**B**) and (**C**) can be found in *Figure 2—source data 2*, and the raw NMR data used to generate (**D**) can be found in *Figure 2—source data 3*.

The online version of this article includes the following source data and figure supplement(s) for figure 2:

**Source data 1.** This folder contains the unedited *Figure 2* and the raw circular dichroism (CD) spectra data shown in *Figure 2A*.

**Source data 2.** This folder contains the raw 2D nuclear magnetic resonance (NMR) spectra data shown in *Figure 2B/C*, which can be opened by commonly used NMR software, including NMRpipe (for processing) and Sparky (for visualization).

**Source data 3.** This folder contains the raw HNCO, HNCA, and HNCOCA nuclear magnetic resonance (NMR) spectra data used to generate the deviation plots in *Figure 2D* and *Figure 2—figure supplement 1F/G*.

**Figure supplement 1.** HPO-30 intracellular domain (ICD) is largely disordered.

**Figure supplement 1—source data 1.** This folder contains the raw circular dichroism (CD) spectra data shown in *Figure 2—figure supplement 1A*.

**Figure supplement 1—source data 2.** This folder contains the unedited *Figure 2—figure supplement 1* and raw gel images used in *Figure 2— figure supplement 1*.

**Figure supplement 1—source data 3.** This folder contains the raw nuclear magnetic resonance (NMR) spectra data shown in *Figure 2—figure supplement 1D/E*, which can be opened by commonly used NMR software, including NMRpipe (for processing) and Sparky (for visualization).

CD spectroscopy analysis and suggested that the ICD is predominantly disordered in solution. Fusing the ICD to two different solubility tags, including the monomeric 56 a.a. tag GB1 (B1 domain of streptococcal protein G, commonly used in NMR) (*Zhou and Wagner, 2010*) and the dimeric GST-tag, did not cause any significant chemical shift changes to the ICD crosspeaks (*Figure 2C*, *Figure 2—figure supplement 1D and E*), suggesting that solubility tags and dimerization do not promote ICD folding. Moreover, the NMR spectra were nearly identical at different protein concentrations (50–700 µM) and temperatures (283–298K), suggesting that the ICD alone does not undergo concentration-dependent structural changes (data not shown).

While the 2D $^1$H-$^{15}$N NMR spectroscopy can provide an overall assessment of the degree of structural disorder based on the magnitude of the $^1$H chemical shift dispersion, a precise characterization of secondary structure content requires measuring the chemical shifts values of individual backbone nuclei. $^{13}$C chemical shift values measured for carbons alpha are particularly strong indicators of

secondary structure motifs. We therefore collected a set of conventional 3D triple resonance experiments (HNCA, HNCO, and HN(CO)CA) and were able to assign the backbone chemical shifts ($^{1}$H, $^{15}$N, and $^{13}$C) of 41 out of 51 a.a. of the ICD (*Figure 2—figure supplement 1F*). The overall measured $^{13}$Cα chemical shifts show little deviation (<1 ppm) from idealized random coil chemical shifts (*Figure 2D*; *Kjaergaard et al., 2011a*; *Kjaergaard and Poulsen, 2011b*), indicating that the ICD is predominantly disordered in solution, consistent with the MD simulations, the CD data, and the initial assessment based on the 2D NMR spectra. However, we observed small but consistent positive secondary Cα chemical shifts, suggesting the presence of a transient (~25–30%) alpha-helical motif in the same region that was predicted to form an alpha helix in many of the ab initio predictions (*Figure 2D*). Although we could not obtain adequate signals for Cβ atoms, the chemical shift of CO was consistent with the Cα chemical shift data, further confirming that the same region of the ICD contained ~25–30% helical motif (*Figure 2—figure supplement 1G*).

Together, our data suggest that the HPO-30 ICD is predominantly disordered in solution, although it does contain some local secondary structures that could potentially fold into different conformations. The exact nature of these conformations is unclear, and it remains challenging to determine whether the entire ICD binds to the WRC as an extended linear peptide or as a compact folded structure.

## HPO-30 ICD requires dimerization to bind WRC

During our investigation of the interaction between the HPO-30 ICD and the WRC, we noticed that GST-tagged ICD exhibited robust binding to WRC, whereas MBP (maltose binding protein)-tagged ICD showed much weaker binding (*Figure 3A*). Note that both constructs contained extensive flexible linkers of at least 16 a.a. between the tag and the ICD to prevent steric hindrance from the tag. Given that GST is a constitutive dimer while MBP is a monomer, we hypothesized that efficient binding of the HPO-30 ICD to WRC may require dimerization.

We explored a variety of strategies to test this hypothesis. We first developed a new WRC construct in which a dual MBP-tag, (MBP)$_2$, was tethered to the WRC. This (MBP)$_2$-WRC would allow us to immobilize the WRC on amylose beads to pull down different monomeric vs. dimeric forms of HPO-30 ICD. Out of three different designs of (MBP)$_2$-WRC, only the one with an (MBP)$_2$ tag tethered to the C-terminus of Abi2 through sortase-mediated protein ligation (*Chen et al., 2011*) effectively pulled down GST-HPO-30 ICD (herein referred to as WRC:Abi2-(MBP)$_2$, *Figure 3—figure supplement 1B*), while the other two versions substantially interfered with the interaction. One construct, WRC:(MBP)$_2$-HSPC300, has an (MBP)$_2$ tag fused to the N-terminus of HSPC300 which was previously used to immobilize the WRC and pull down WIRS-containing proteins (*Chen et al., 2014b*; *Figure 3—figure supplement 1B*, lane 6 vs 8), and the other, WRC:(MBP)$_2$-Abi2, has (MBP)$_2$ fused to the N-terminus of Abi2 (*Figure 3—figure supplement 1A*). Thus, it is possible that the regions near the N-terminus of HSPC300 or Abi2 are close to the HPO-30 ICD binding surface (but not the WIRS binding surface).

Using WRC:Abi2-(MBP)$_2$, we compared the binding of ICDs that carry various monomeric vs. dimeric tags. We observed that the dimeric form of GB1-ICD (referred to as GB1-CC-ICD, in which we inserted a homodimeric coiled-coil sequence between GB1 and the ICD; *O'Shea et al., 1991*) exhibited much stronger binding than the monomeric GB1-ICD (*Figure 3B*). To rule out the possibility that the tag sequences, rather than dimerization, influenced protein binding, we introduced chemically inducible dimerization (CID) tags, FKBP and FRB, to the ICD (with the GB1 tag at the N-terminus) (*Banaszynski et al., 2005*; *Figure 3C*). In the presence of the dimerizing agent rapamycin, GB1-FKBP-ICD and GB1-FRB-ICD could form a tight heterodimer, as confirmed by size exclusion chromatography in the presence and absence of rapamycin (*Figure 3—figure supplement 1C–E*). Consistent with results from the constitutive dimeric tags, the addition of rapamycin to dimerize GB1-FKBP/FRB-ICD (F/F-ICD for short hereafter) clearly promoted the binding of HPO-30 ICD (*Figure 3C*, lanes 2 vs 3). In fact, we observed robust binding of all dimeric constructs we tested, including GST-ICD, ICD-GST, and DLC8-ICD (DLC8 is a constitutive dimer from the dynein light chain) (*Wang et al., 2003*). In contrast, all monomeric constructs including GB1-ICD, sumo-ICD, and mEGFP-ICD exhibited very weak, if any, binding (*Figure 3—figure supplement 2A,B*).

ICD binding relies on the dimerization tags to bring two copies of ICD in close proximity, because a heterodimer containing only one copy of ICD (formed by mixing GB1-FKBP-ICD with GB1-FRB tag) was unable to support binding (*Figure 3D*, lanes 5–6). Similarly, a heterodimer containing one WT

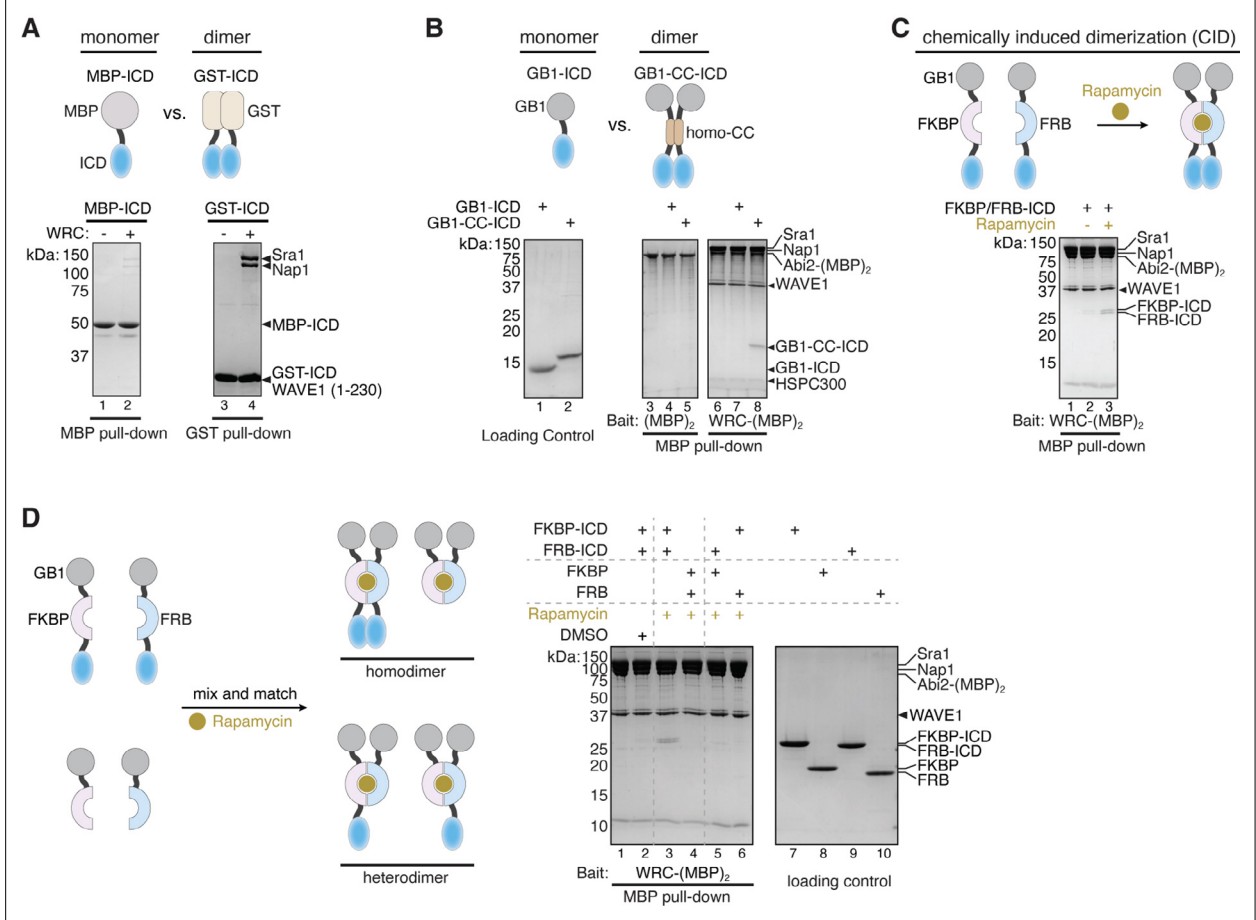

**Figure 3.** Dimerization tags enhance HPO-30 intracellular domain (ICD) binding to the WAVE regulatory complex (WRC). (**A–C**) Top: Schematic of monomeric and dimeric HPO-30 ICD constructs used in pull-downs. Bottom: Coomassie blue-stained SDS-PAGE gels showing (**A**) MBP-ICD or GST-ICD (200 pmol) pulling down WRC$^{230\Delta WCA}$ (200 pmol), (**B**) WRC:Abi2-(MBP)$_2$ (60 pmol) pulling down GB1-ICD or GB1-CC-ICD (600 pmol), and (**C**) WRC:Abi2-(MBP)$_2$ (60 pmol) pulling down FKBP/FRB-ICD in the absence or presence of rapamycin. Where it is noted, 5 µM rapamycin (or equal volume of DMSO as a negative control) was used to induce dimerization. For clarity, GB1-FKBP-ICD and GB1-FRB-ICD were referred to as FKBP-ICD and FRB-ICD, respectively. (**D**) Schematic of the assembly of homo- and heterodimers of HPO-30 ICD (left) and Coomassie blue-stained SDS-PAGE gels (right) showing WRC:Abi2-(MBP)$_2$ (60 pmol) pulling down indicated homo- and heterodimers. In homo- and heterodimers, 1200 pmol of wild-type (WT) ICD monomeric unit was included to ensure the total number of WT ICD molecules remained the same. The raw gel images and unedited figure can be found in *Figure 3—source data 1*.

The online version of this article includes the following source data and figure supplement(s) for figure 3:

**Source data 1.** This folder contains the unedited *Figure 3* and raw gel images used in *Figure 3*.

**Figure supplement 1.** Protein quality control for dimerization experiments.

**Figure supplement 1—source data 1.** This folder contains the unedited *Figure 3—figure supplement 1* and raw gel images used in *Figure 3—figure supplement 1*.

**Figure supplement 1—source data 2.** This folder contains the raw gel filtration chromatogram data shown in *Figure 3—figure supplement 1C/E*.

**Figure supplement 2.** HPO-30 intracellular domain (ICD) binding to WAVE regulatory complex (WRC) requires dimerization tags.

**Figure supplement 2—source data 1.** This folder contains the unedited *Figure 3—figure supplement 2* and raw gel images used in *Figure 3—figure supplement 2*.

ICD and one mutant ICD (alanine Δ5 mutation in *Figure 1*) had weakened binding (*Figure 3—figure supplement 2F*). It is important to note that when using the heterodimeric ICDs, we maintained the same total monomeric concentration of the WT ICD between heterodimers and homodimers. Thus, the lack of binding from heterodimers was due to the absence of the WT ICD dimer rather than a reduction in WT ICD concentration.

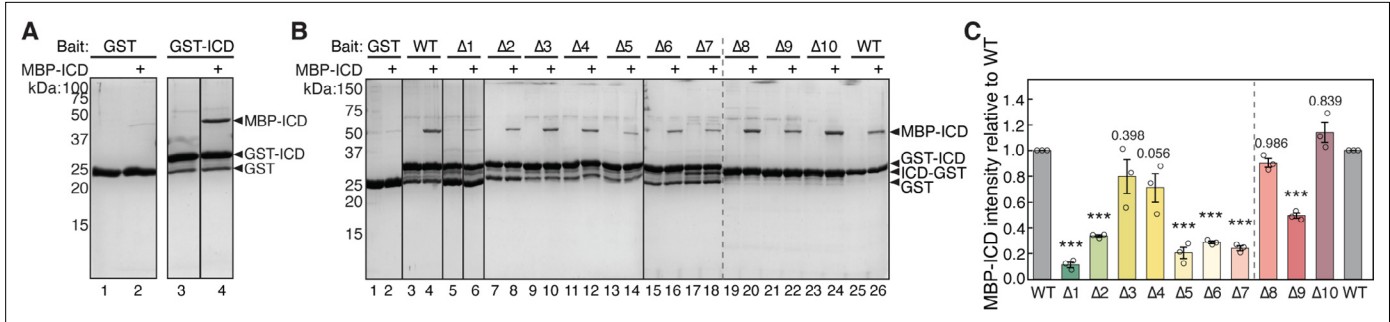

**Figure 4.** HPO-30 intracellular domain (ICD) can self-associate. (**A**) Coomassie blue-stained SDS-PAGE gels showing GST-ICD (300 pmol) pulling down MBP-ICD (6000 pmol). (**B**) Representative Coomassie blue-stained SDS-PAGE gels of three independent experiments showing GST-ICD alanine mutations (shown in *Figure 1A*) (300 pmol) pulling down MBP-ICD (6000 pmol). (**C**) Quantification of (**B**). The MBP-ICD band intensity was used to quantify the pull-down signal. Signals from GST-ICD or ICD-GST pull-downs were normalized to corresponding wild-type (WT) ICD (lanes 4 and 26, respectively). Error bars represent standard error, *p<0.05, **p<0.005, ***p<0.001 from ANOVA on ranks. n = 3. The raw gel images and unedited figure can be found in *Figure 4—source data 1*. The raw data for (**C**) can be found in *Figure 4—source data 2*.

The online version of this article includes the following source data and figure supplement(s) for figure 4:

**Source data 1.** This folder contains the unedited *Figure 4* and raw gel images used in *Figure 4*.

**Source data 2.** This folder contains the raw data and statistical analysis shown in *Figure 4C*.

**Figure supplement 1.** HPO-30 intracellular domain (ICD) can interact with itself.

**Figure supplement 1—source data 1.** This folder contains the unedited *Figure 4—figure supplement 1* and raw gel images used in *Figure 4— figure supplement 1*.

**Figure supplement 2.** Tryptophan fluorescence cannot be used to monitor the self-association of HPO-30 intracellular domain (ICD).

**Figure supplement 2—source data 1.** This folder contains the unedited *Figure 4—figure supplement 2* and raw gel images used in *Figure 4— figure supplement 2*.

**Figure supplement 2—source data 2.** This folder contains the raw data used in *Figure 4—figure supplement 2A, B, D, and E*.

**Figure supplement 3.** Structural predictions of HPO-30 intracellular domain (ICD) self-association.

**Figure supplement 3—source data 1.** This folder contains the unedited *Figure 4—figure supplement 3* and two Pymol scene files containing all predicted structural models shown in *Figure 4—figure supplement 3* aligned to one another.

**Figure supplement 3—source data 2.** This folder contains individual Pymol scene files of the starting and final conformations for all models used in molecular dynamics (MD) simulations and root mean square deviation (RMSD) values for each simulation.

Various competition assays further support the observation that only dimeric form of the ICD can effectively bind to WRC. For example, even when added in 5000-fold excess, the chemically synthe-sized HPO-30 ICD peptide could not compete off GST-ICD binding (*Figure 3—figure supplement 2C*). In contrast, both F/F-ICD and DLC8-ICD binding to the WRC could be effectively competed off by GST-ICD in a dose-dependent manner (*Figure 3—figure supplement 2D*). Moreover, F/F-ICD was able to compete off GST-ICD binding to WRC only in the presence of rapamycin, but not in its absence (*Figure 3—figure supplement 2E*, lanes 4 and 5). Taken together, our data strongly suggest that dimerization of the HPO-30 ICD is critical for its binding to the WRC.

## HPO-30 ICD can self-associate

How do dimeric tags promote ICD binding to WRC? We hypothesized that dimeric tags facilitate ICD dimerization, and only dimerized ICD can bind to WRC. To test this hypothesis, we first examined whether the ICD could interact with another copy of ICD, which would indicate the potential for ICD dimerization. Indeed, we found GST-ICD effectively pulled down MBP-ICD, indicating self-interaction (*Figure 4A*). This is consistent with our co-immunoprecipitation assay, which showed that HA-tagged full-length HPO-30 retained Myc-tagged full-length HPO-30 when co-expressed in *Drosophila* S2 cells (*Figure 4—figure supplement 1B*). The ICD-ICD interaction was readily disrupted by increased salt concentration (not shown) or higher pH (*Figure 4—figure supplement 1A*), suggesting that the inter-action likely involves electrostatic interactions. We further asked which region in the HPO-30 ICD was important for the ICD-ICD interaction. Using the same alanine scan mutants as in *Figure 1A*, we found that mutations in various regions impaired the ICD-ICD interaction, especially the N-terminal (Δ1 and

Δ2) and the central regions (Δ5, Δ6, and Δ7) of the ICD (*Figure 4B*). It is remarkable that Δ5, Δ6, and Δ7 sequences are also crucial for binding to WRC (*Figure 1B*), suggesting that intracellular domain (ICD) dimerization is closely linked to the WRC interaction.

Due to technical challenges, we were unable to measure the binding affinity of the ICD-ICD interaction using intrinsic tryptophan fluorescence (*Figure 4—figure supplement 2A*). Our structural predictions using AlphaFold Multimer generated several different models, depending on the context of the ICD sequence used (*Figure 4—figure supplement 3A, B*; *Evans et al., 2022*; *Jumper et al., 2021*). The isolated ICD was consistently predicted to form a homodimer (*Figure 4—figure supplement 3A*). In contrast, the full-length HPO-30 was predicted to form two distinct dimers, one mediated by the transmembrane helix 2 (H2) and the beta strand 4 (b4) in the extracellular domain, and the other mediated by the transmembrane helix 3 (H3) and the first helix of the ICD (*Figure 4—figure supplement 3B*). In both conformations, the ICD was predicted to form a long helix, without the two beta strands predicted for the monomeric ICD (*Figure 1A*; *Figure 1—figure supplement 2*; *Figure 4—figure supplement 3A*). All-atom MD simulations conducted on the dimeric structural models indicate that the dimer conformations are not stable over the course of 200 ns, suggesting that the limited helical interface between the two ICDs is not sufficient to form a stable complex (*Figure 4—figure supplement 3C*).

In summary, our data suggest that HPO-30 ICD requires dimerization to interact with the WRC. However, before to binding the WRC, the dimerized ICD may not have a stable quaternary structure.

## HPO-30 ICD inhibits actin polymerization, but promotes Rac1-mediated WRC activation

Rac1 GTPase is the canonical activator of the WRC, and various WIRS-containing receptors can further fine-tune WRC activation by Rac1 (*Chen et al., 2014b*), either positively or negatively. We wondered how HPO-30 ICD binding could influence WRC activity in promoting Arp2/3-mediated actin polymerization (*Cooper et al., 1983*; *Doolittle et al., 2013a*; *Kouyama and Mihashi, 1981*). For this, we compared the effect of monomeric and dimeric ICD on WRC activity using an optimized buffer in the pyrene-actin polymerization assay (*Figure 5—figure supplement 1A*). The CID FKBP/FRB-ICD constructs (shown in *Figure 3C*) allowed us to directly compare the monomeric vs. dimeric ICD by switching between DMSO and rapamycin, instead of using different tags (*Figure 3—figure supplement 1C, D*). Importantly, rapamycin alone did not have any effect on actin polymerization (*Figure 5—figure supplement 1B*), allowing us to focus on the effect of dimerizing HPO-30 ICD. Unexpectedly, we observed that HPO-30 ICD strongly inhibits Rac1-WRC-mediated actin polymerization (*Figure 5A*, blue vs. purple curves), with the dimerized ICD exhibiting a more potent effect (*Figure 5A*, solid vs. dashed curves). This effect was dose-dependent, with 10 µM ICD having a stronger effect than 5 µM ICD (*Figure 5—figure supplement 1D*). In contrast, the dimerized FKBP/FRB tag at the highest concentration only had a mild effect (*Figure 5—figure supplement 1D*, gray curve). Importantly, the presence of ICD did not non-specifically quench pyrene fluorescence from either G-actin or F-actin (*Figure 5—figure supplement 1C*). Therefore, the observed inhibition was due to HPO-30 ICD preventing actin polymerization.

Interestingly, we noticed that the ICD inhibited actin polymerization to a level lower than the basal level of actin polymerization from Arp2/3 alone, indicating that the inhibitory effect was likely not through WRC. This is supported by our observation that the ICD similarly inhibited actin polymerization induced by the isolated, constitutively active WCA peptide, with the dimerized ICD again showing more potent inhibition (*Figure 5B*). In fact, the HPO-30 ICD was able to inhibit spontaneous actin polymerization in the absence of any activator, including the Arp2/3 complex (*Figure 5C*, *Figure 5—figure supplement 1E*). Therefore, our data suggest that the inhibitory effect of the ICD is likely a direct effect on actin, rather than through WRC, Rac1, WCA, or Arp2/3.

The strong, WRC-independent inhibitory effect of the ICD on actin polymerization made it difficult to directly examine how ICD binding influences WRC activity. To overcome this challenge, we reduced the ICD concentration to 1 µM to minimize its inhibitory effect. At this concentration, we did not observe HPO-30 ICD activating WRC alone (*Figure 5D*). Previous studies on the WIRS-containing ICD from the cell adhesion membrane protein protocadherin 10 (PCDH10) found that the PCDH10 ICD itself could not activate WRC but could enhance WRC activation induced by low concentrations of Rac1 (*Chen et al., 2014b*). We tested if HPO-30 ICD might act similarly to PCDH10, enhancing

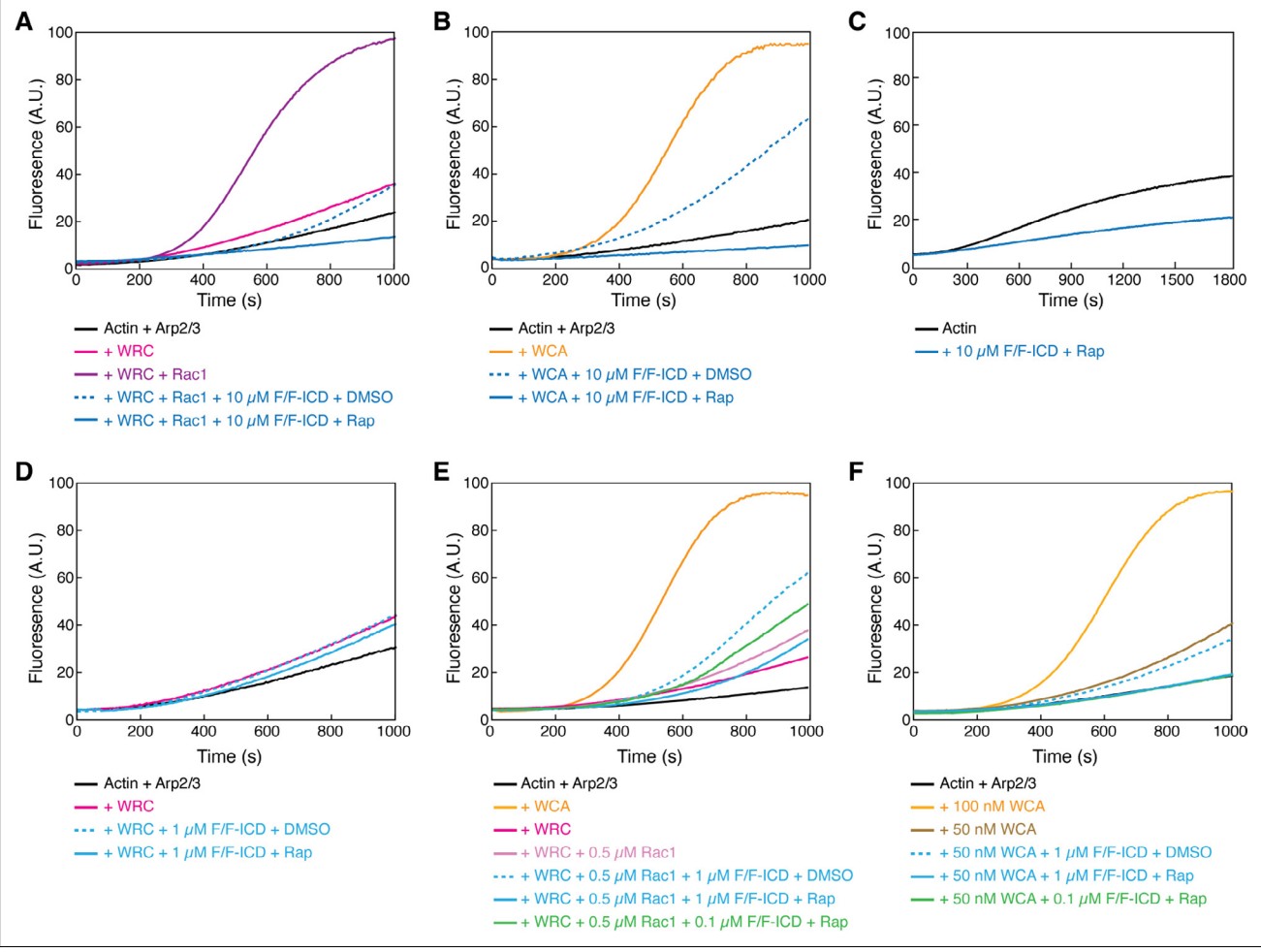

**Figure 5.** HPO-30 intracellular domain (ICD) inhibits actin polymerization but promotes Rac1-WAVE regulatory complex (WRC) activation. (**A–F**) Pyrene-actin polymerization assays of indicated conditions. Reactions contain 2 µM actin (10% pyrene-labeled) (**A–B, D–F**), 10 nM Arp2/3 complex, 100 nM WRC230WCA or isolated WCA, Rac1QP, indicated concentrations of equimolar GB1-FKBP-ICD and GB1-FRB-ICD, and 5 µM rapamycin or equal volume of DMSO. (**C**) contains 4 µM actin (5% pyrene-labeled), indicated concentration of equimolar GB1-FKBP-ICD and GB1-FRB-ICD, and 5 µM rapamycin. The raw data used to generate these curves can be found in *Figure 5—source data 1*.

The online version of this article includes the following source data and figure supplement(s) for figure 5:

**Source data 1.** This folder contains the unedited *Figure 5* and the raw pyrene-actin polymerization data shown in *Figure 5A–F*.

**Figure supplement 1.** HPO-30 intracellular domain (ICD) inhibits actin polymerization.

**Figure supplement 1—source data 1.** This folder contains the unedited *Figure 5—figure supplement 1* and raw gel images used in *Figure 5—figure supplement 1*.

**Figure supplement 1—source data 2.** This folder contains the raw pyrene-actin polymerization data shown in *Figure 5—figure supplement 1B–F* and the gel quantification data shown in *Figure 5—figure supplement 1F*, right.

activity when WRC is mildly activated by Rac1. Interestingly, in this condition, we found that 1 µM monomeric ICD slightly, but robustly, increased Rac1-WRC-mediated actin polymerization (*Figure 5E*, dashed blue vs. solid light pink curves). Since only dimerized ICD can readily bind WRC, we speculated that the enhanced activity was due to a small fraction of dimeric ICD existing in equilibrium with the monomeric form (*Figure 4*, *Figure 4—figure supplement 1*). To test this hypothesis (while avoiding the dominant inhibitory effect of rapamycin-dimerized ICD observed even at 1 µM, *Figure 5E*, solid blue curve), we reduced the dimerized ICD concentration to 0.1 µM. Consistent with our hypothesis, at this low concentration, the dimerized ICD similarly promoted Rac1-WRC-medicated actin polymerization (*Figure 5E*, solid green curve). This activating effect was specific to Rac1-activated WRC, as free WCA could not be further activated by dimerized ICD (*Figure 5F*). Although we could not test

higher concentrations of ICD due to its strong inhibitory effect on actin polymerization, low concentrations of dimeric ICD could enhance Rac1-WRC activity, suggesting ICD binding cooperates with Rac1 to promote WRC activation.

The cooperativity between Rac1 and the HPO-30 ICD in activating WRC suggests that the ICD has a preference for binding to activated WRC. Supporting this hypothesis, our EPD assay showed that GST-ICD pulled down more WRC in the presence of saturating concentrations of Rac1 (which was manifested by less WRC remaining in the pull-down supernatant, *Figure 5—figure supplement 1G*, lane 2 vs 4; *Yang et al., 2022*). Taken together, our data demonstrate that, while the HPO-30 ICD cannot activate the WRC by itself, it can synergistically cooperate with Rac1 to promote WRC activation, similar to the WIRS receptor PCDH10 (*Chen et al., 2014b*). Our unexpected finding that the HPO-30 ICD strongly inhibits actin polymerization suggests that the ICD directly interacts with actin.

## HPO-30 ICD binds to F-actin and inhibits actin depolymerization similar to CapZ

The strong inhibitory effect of the HPO-30 ICD on actin polymerization suggests the ICD could directly bind and sequester monomeric G-actin. However, we could not detect binding between GST-ICD or dimerized F/F-ICD and G-actin (*Figure 6—figure supplement 1A, B*). We next wondered if the ICD inhibited actin polymerization by binding to F-actin. In our F-actin co-pelleting assay (*Heier et al., 2017*), we found that, compared to BSA, F/F tag, or the capping protein CapZ (which only binds to the barbed ends of F-actin), F/F-ICD more robustly bound to F-actin (*Figure 6A and B*). Interestingly, both the monomeric and dimeric F/F-ICD showed significant binding to F-actin (*Figure 6A and B*, blue, +DMSO vs. +Rap), although the dimeric ICD showed significantly stronger binding. This suggests that the monomeric ICD, while not as effective in inhibiting actin polymerization as the dimeric form, is also capable of binding to actin filaments. An equilibrium actin pelleting assay suggested a $K_D$ of ~1.69 μM for the binding between the dimerized ICD and F-actin (*Figure 6—figure supplement 1C*; *Avery et al., 2017*).

To identify the critical region(s) of the ICD required for its interaction with actin filaments, we introduced the same alanine scan mutations used in *Figure 1* into both SNAP-tagged FKBP-ICD and FRB-ICD constructs, which we used for all subsequent actin-related experiments. Interestingly, we found that the same regions crucial for ICD's interaction with the WRC (*Figure 1B*) and with itself (*Figure 4B and C*), including Δ1, Δ5, Δ6, and Δ7, were also the most essential for binding to F-actin (*Figure 6—figure supplement 1D and E*).

The above data strongly suggest that the HPO-30 ICD inhibits actin polymerization by directly binding to F-actin. Proteins with similar activity include capping proteins, such as CapZ. Capping proteins bind to the barbed end of actin filaments to impede both actin polymerization and depolymerization from the barbed end (*Caldwell et al., 1989*; *Weeds and Maciver, 1993*). To test whether the HPO-30 ICD may function as a capping protein, we used a depolymerization assay that is commonly utilized to measure CapZ activity (*Caldwell et al., 1989*; *Cooper and Pollard, 1985*). We found that the ICD, like CapZ, inhibited actin depolymerization (*Figure 6C and D*). This inhibitory activity was both concentration- and dimerization-dependent, suggesting that dimerization is critical for this capping-like activity. This aligns with our observation that dimeric ICD more potently inhibited actin polymerization than monomeric ICD (*Figure 5*, *Figure 5—figure supplement 1E*). To determine which region(s) of the ICD are important for this capping-like activity, we tested the same alanine scan mutations used in *Figure 1* and *Figure 6—figure supplement 1D and E*. As anticipated, the regions most critical for the depolymerization activity of the ICD were the same as those crucial for binding to actin filaments (*Figure 6—figure supplement 1F–G*). Collectively, our results suggest that the HPO-30 ICD behaves analogously to capping proteins in obstructing both depolymerization and polymerization of F-actin (*Figure 5*, *Figure 6*).

## HPO-30 ICD binds to both the side and barbed end of actin filaments

In the above bulk solution assays, the binding of the ICD to F-actin was not dependent on dimerization (*Figure 6A and B*), but its inhibition of F-actin polymerization and depolymerization was dependent on dimerization (*Figures 5A–C, 6C and D*). This suggests that the HPO-30 ICD may have two distinct activities on actin filaments, with only the dimerization-dependent activity resembling that of CapZ. This hypothesis could explain why more ICD co-pelleted with F-actin than CapZ did (*Figure 6A*

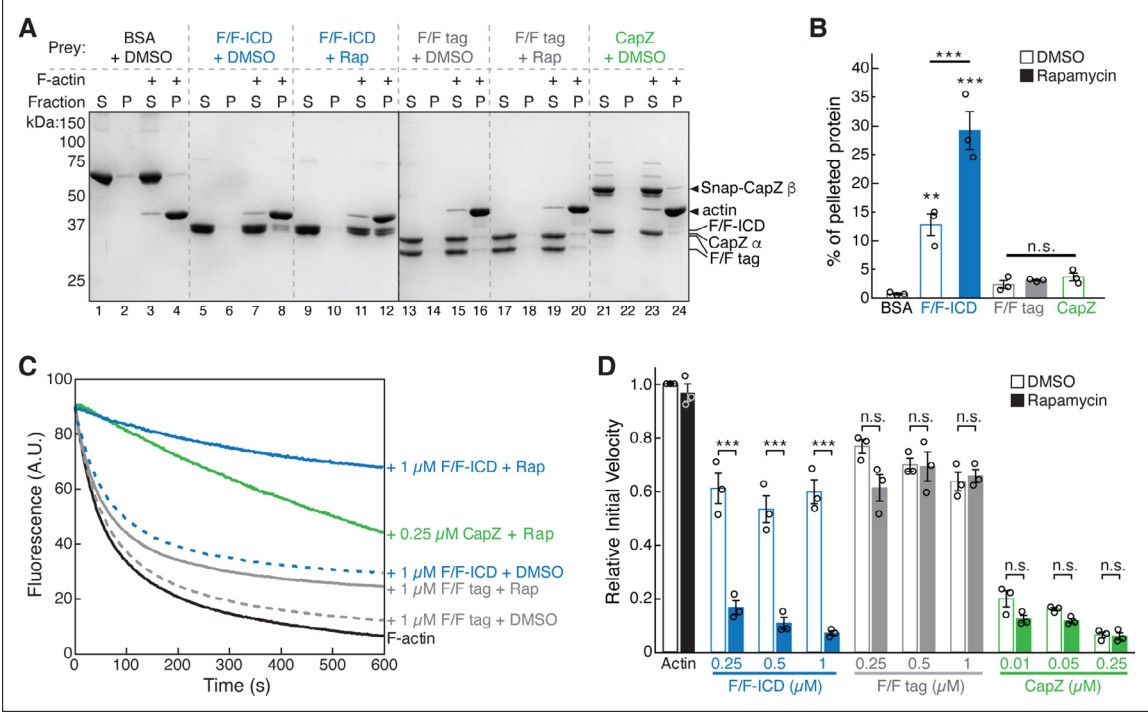

**Figure 6.** HPO-30 intracellular domain (ICD) binds to F-actin and inhibits actin depolymerization. (**A**) Representative Coomassie blue-stained SDS-PAGE gels showing F-actin co-pelleting assays of indicated F/F-ICD and CapZ in the presence or absence of 5 µM rapamycin (or equal volume of DMSO). S: supernatant, P: pellet. (**B**) Quantification of (**A**) from three independent repeats, showing percentage of proteins in the pellet. Bars represent standard error, \*\*p<0.005, \*\*\*p<0.001, ANOVA with Tukey test. n = 3. (**C**) Representative F-actin depolymerization assay fluorescence curves of indicated conditions. Each reaction contained 5 µM pre-polymerized actin (70% pyrene labeled), diluted 20-fold into depolymerization buffer containing indicated proteins in the presence or absence of 5 µM rapamycin (or equal volume of DMSO). (**D**) Quantification of the initial velocity of fluorescence curves shown in (**C**). Initial velocity was normalized to F-actin alone in the presence of DMSO. Error bars represent standard error, n=3 independent repeats, \*\*\*p<0.001, ANOVA with Tukey test. n = 3. The raw gel images and unedited figure can be found in *Figure 6—source data 1*. The raw data used to create (**B**, **C**, **D**) can be found in *Figure 6—source data 2*.

The online version of this article includes the following video, source data, and figure supplement(s) for figure 6:

**Source data 1.** This folder contains the unedited *Figure 6* and raw gel images used in *Figure 6*.

**Source data 2.** This folder contains the raw data and statistical analysis shown in *Figure 6B, C, and D*.

**Figure supplement 1.** HPO-30 intracellular domain (ICD) binds to F-actin, but not G-actin, and inhibits F-actin depolymerization.

**Figure supplement 1—source data 1.** This folder contains the unedited *Figure 6—figure supplement 1* and raw gel images used in *Figure 6—figure supplement 1*.

**Figure supplement 1—source data 2.** This folder contains the raw data and statistical analysis shown in *Figure 6—figure supplement 1C, E, F, G*.

**Figure 6—video 1.** CapZ capping event example 1.
https://elifesciences.org/articles/88492/figures#fig6video1

**Figure 6—video 2.** CapZ capping event example 2.
https://elifesciences.org/articles/88492/figures#fig6video2

**Figure 6—video 3.** HPO-30 intracellular domain (ICD) side binding event example 1.

https://elifesciences.org/articles/88492/figures#fig6video3

**Figure 6—video 4.** HPO-30 intracellular domain (ICD) side binding event example 2.
https://elifesciences.org/articles/88492/figures#fig6video4

**Figure 6—video 5.** Actin elongation rate comparison.
https://elifesciences.org/articles/88492/figures#fig6video5

*and B*), as CapZ only binds to the barbed end of actin filaments (albeit with high affinity) (*Caldwell et al., 1989*), while the ICD may bind to both the barbed end (in a dimerization-dependent manner) and other surfaces of the actin filaments (in a dimerization-independent manner).

To understand how the HPO-30 ICD binds F-actin, we used single-molecule total internal reflection fluorescence (smTIRF) microscopy to directly visualize the interaction. For this assay, we first attached a SNAP tag to the N-terminus of various FKBP/FRB constructs or the CapZ β-subunit, and then labeled the SNAP tag using different SNAP-Surface Alexa Fluor dyes (hereafter referred to as AF followed by the corresponding excitation wavelength). Actin was directly labeled with AF647 using previously established methods (*Hansen et al., 2013*). We verified that the SNAP-tagged F/F-ICDs bound to the WRC in a rapamycin-dependent manner and to a level similar to the GB1-tagged F/F-ICD, suggesting that the SNAP tag did not affect ICD function (*Figure 7—figure supplement 1A*, lanes 1–4). The labeled CapZ and F/F-ICD proteins exhibited capping activity comparable to their counterparts used in bulk solution assays (except for the AF647-labeled F/F-ICD, which was subsequently excluded from smTIRF assays described below) (*Figure 7—figure supplement 1B and C*).

In the smTIRF experiments, SNAP$^{AF488}$-CapZ (at 5 nM) was found to stably bind to the barbed end of filaments, and the binding stopped filament growth (*Figure 7A*, top; *Figure 7* green; *Figure 6—videos 1 and 2*), consistent with the high affinity and very slow off-rate of CapZ at the barbed end shown in previous studies (*Caldwell et al., 1989*). In contrast, SNAP$^{AF546}$-F/F-ICD (15 nM, with eight-fold molar excess of rapamycin) mainly bound to the side of actin filaments, instead of the barbed end (*Figure 7A*, bottom; *Figure 7B*; *Figure 6—videos 3–5*) (see below). While we occasionally observed an ICD appearing to bind to the barbed end of a filament (see *Figure 7A*, bottom, for an example, at time 39.0 s), the filament continued to grow beyond the bound ICD, leaving the ICD bound to the same location on the filament. We could conclude that the ICD molecule bound to the filament and not the PEG-coated surface, because it moved together with the filament (*Figure 6—videos 3 and 4*). By using stringent criteria to prevent misidentification of non-specific background signals as binding events, we found that 33% of filaments had at least one binding event for SNAP-F/F-ICD over the 15 min duration of the experiment, which was significantly higher than the number of events for the SNAP-F/F tag under identical conditions (*Figure 6B*).

We found it difficult to increase the concentration of fluorescently labeled ICD to observe more barbed end binding events, as this resulted in high background in smTIRF experiments. Therefore, we used higher concentrations of unlabeled ICD to see if it could inhibit actin filament growth as shown in the bulk solution assays in *Figure 6C and D* (*Figure 7C*; *Figure 6—video 5*). We found that 1 µM dimerized F/F-ICD significantly decreased the actin elongation rate from 2.3±0.5 subunits/s to 1.5±0.3 subunits/s (*Kuhn and Pollard, 2005*; *Figure 7C*, black vs. solid blue). The same concentration of monomeric F/F-ICD also slowed down actin elongation, but to a lesser extent (2.0±0.4 subunits/s) (*Figure 7C*, open blue). This could be due to either the side binding activity of ICD or the presence of dimeric ICD in equilibrium with monomeric ICD, which could bind to the barbed end. Importantly, dimerized F/F tag did not significantly affect actin polymerization (2.1±0.4 subunits/s) (*Figure 7C*, gray). Unfortunately, we could not increase the unlabeled ICD concentration further to observe stronger inhibitory effects due to limitations in the volume of high-concentration protein stock that we could add to the microfluid chamber. In contrast, CapZ strongly inhibited actin polymerization, even at 5 nM (0.3±0.1 subunits/s). Further examination of the kymographs of individual actin filaments revealed that in the absence of ICD or CapZ, filaments typically grew at a uniform rate. In the presence of dimeric F/F-ICD, filaments grew more slowly and with occasional pausing (*Figure 7E*; *Figure 7—figure supplement 2A and B*). In the presence of CapZ, filaments exhibited almost no growth most of the time over the course of the video (*Figure 7E*; *Figure 7—figure supplement 2C*). Taken together, our data suggest that HPO-30 ICD and CapZ have similar inhibitory effects on actin filament elongation, although at the same concentration, the ICD exhibits less potent activity.

In the above smTIRF experiments, we encountered several technical challenges that prevented us from observing stable binding of the ICD to the barbed end, in contrast to the stable binding observed with CapZ. These challenges include limited concentrations of fluorophore-labeled ICD (nM, instead of µM that we could use in bulk solution assays), potential low affinity and fast off-rate of the ICD binding to the barbed end, difficulty in distinguishing barbed end binding from side binding due to light diffraction limit, lower than expected concentration of dimeric ICD (the dissociation constant of FKBP-rapamycin binding to FRB is ~12 nM; *Banaszynski et al., 2005*), and fast binding/

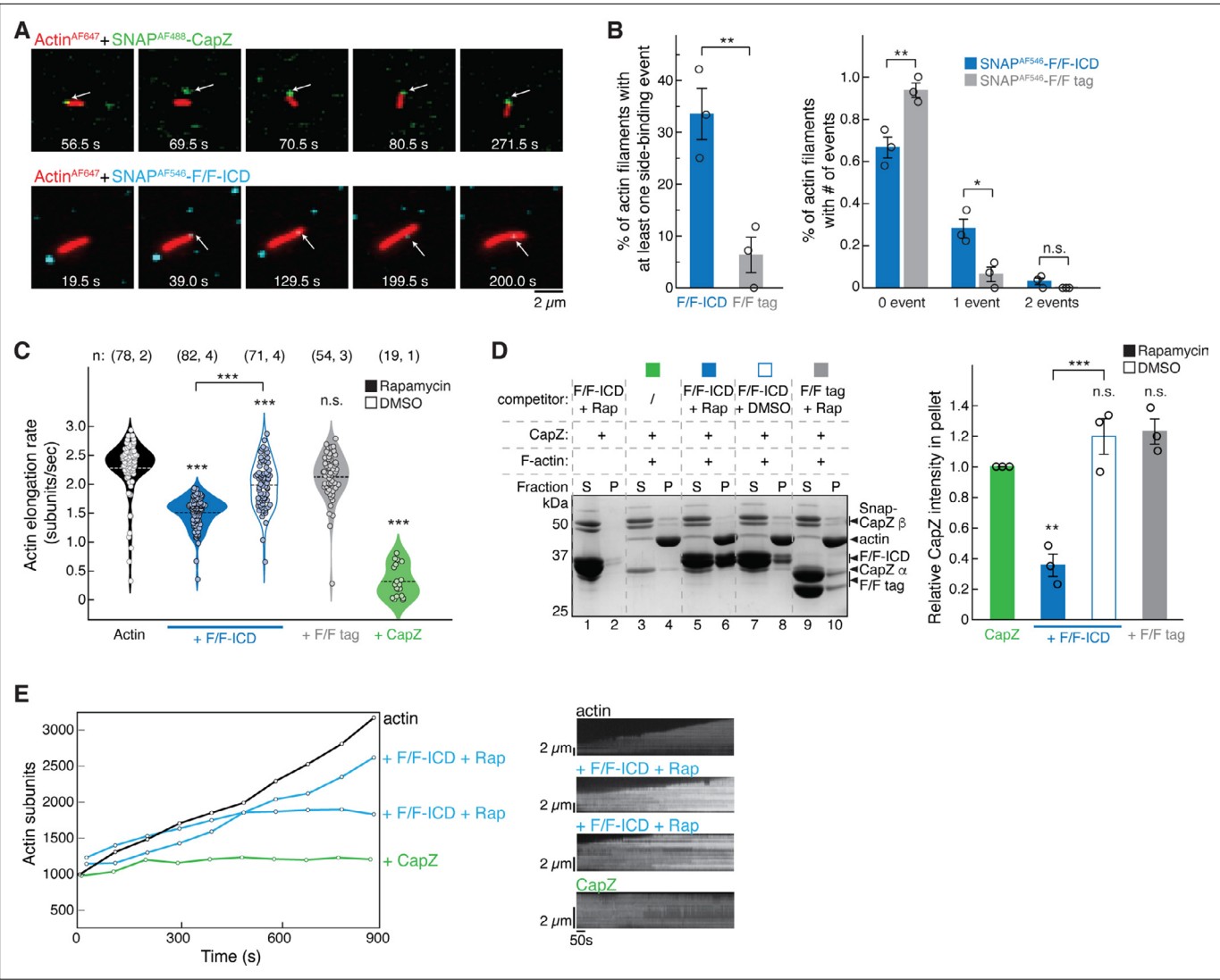

**Figure 7.** HPO-30 intracellular domain (ICD) binds both the side and barbed end of actin filaments and slows polymerization. (**A**) Examples of time lapse images from single-molecule total internal reflection fluorescence (smTIRF) experiments. Top: A capping event of SNAP^AF488-CapZ (green, 5 nM) and actin^AF647 (red). Bottom: A side binding event of SNAP^AF546-F/F-ICD (cyan, 15 nM) and actin^AF647 (red). Note that in smTIRF time lapses, sometimes a lag was observed between signals at the two channels. For example, in the top row, the filament and SNAP^AF488-CapZ puncta were displaced at 69.5 s, but aligned again at 70.5 s. The lagging was due to image acquisition conditions, where, to prevent photobleaching of AF647, images in the 640 nm channel for actin^AF647 were taken every 20 acquisitions of the 488 nm channel for SNAP^AF488-CapZ (see Materials and methods). (**B**) Quantification of the frequency of side binding events from smTIRF imaging. Left: Comparison of overall side binding events. Right: Comparison of the percentage of filaments with 0, 1, and 2 events. Data were from three independent repeats, at least 15 filaments per video. Bars represent standard error. *p<0.05, **p<0.01, Student's paired t-test. n = 3. (**C**) Violin plot of the elongation rate of actin filaments measured by smTIRF imaging. Total number of filaments pooled from the number of independent repeats for each sample are indicated in the paratheses on top of each column, respectively. ***p<0.001, ANOVA with Tukey test. (**D**) Representative Coomassie blue-stained SDS-PAGE gel (left) and quantification of pelleted CapZ from three independent repeats (right) showing F-actin (2 µM) co-pelleting with CapZ (0.6 µM), with the indicated F/F-ICD or F/F tag as a competitor (5 µM) in the presence or absence of 5 µM rapamycin (or equivalent volume DMSO). S: supernatant, P: pellet. Bars represent standard error, **p<0.01, ***p<0.001, ANOVA with Tukey test. n = 3. (**E**) Left: Representative traces of actin filament length determined manually from individual frames. Filaments were chosen that were the closest to the average rate of elongation, with one additional filament for the ICD reactions with a reduced elongation rate. Right: Kymographs of the filaments displayed on the left. The unedited figure and raw gel images can be found in *Figure 7—source data 1*, the raw data used to create (**B**, **C**, **D**, right, **E**, left) can be found in *Figure 7—source data 2*, and the kymographs can be found in *Figure 7—source data 3*.

The online version of this article includes the following source data and figure supplement(s) for figure 7:

**Source data 1.** This folder contains the unedited *Figure 7* and raw gel images used in *Figure 7*.

**Source data 2.** This folder contains the raw data and statistical analysis shown in *Figure 7B–E*.

*Figure 7 continued on next page*

*Figure 7 continued*

**Source data 3.** This folder contains the kymographs for *Figure 7E*.

**Figure supplement 1.** SNAP tag and fluorophore labeling did not affect HPO-30 intracellular domain (ICD) function.

**Figure supplement 1—source data 1.** This folder contains the unedited *Figure 7—figure supplement 1* and raw gel images used in *Figure 7—figure supplement 1*.

**Figure supplement 1—source data 2.** This folder contains the raw data shown in *Figure 7—figure supplement 1C*.

**Figure supplement 2.** HPO-30 intracellular domain (ICD) affects actin polymerization rate.

**Figure supplement 2—source data 1.** This folder contains the raw data shown in *Figure 7—figure supplement 2*.

**Figure supplement 2—source data 2.** This folder contains the kymographs shown in *Figure 7—figure supplement 2*.

---

dissociation events potentially beyond the data acquisition speed (50 or 100 ms exposure time). To further investigate whether the ICD binds to the barbed end, we used a competition co-pelleting assay to determine if dimeric F/F-ICD could compete with CapZ for binding to the barbed end of F-actin (*Figure 7D*). Our results showed that the dimerized F/F-ICD, when applied at about 10 times the concentration of CapZ, significantly decreased CapZ binding by ~60%. In contrast, neither the monomeric F/F-ICD nor the dimerized F/F tag had an impact on CapZ binding (*Figure 7D*). This result suggests that dimeric ICD binds to the barbed end and can block CapZ binding, consistent with our findings that dimeric ICD is more effective in inhibiting actin polymerization (*Figure 5A and B*) and depolymerization (*Figure 6C and D*).

Combining our single-molecule and bulk solution results, we conclude that HPO-30 ICD monomers preferentially bind to the side of actin filaments, while dimers can bind to both the sides and the barbed end of filaments. Only the barbed end binding by dimeric ICD can inhibit actin filament growth in a manner similar to CapZ. These findings reveal a surprising function of the HPO-30 ICD in directly regulating actin cytoskeletal remodeling.

## Discussion

HPO-30 is a claudin-like membrane protein with a crucial role in higher order dendrite branching in *C. elegans* PVD neurons (*Smith et al., 2013*; *Tang et al., 2019*; *Zou et al., 2018*; *Zou et al., 2015*). At branching sites, HPO-30 forms a co-receptor with the cell adhesion transmembrane protein DMA-1, which together transduce extracellular signals (from the epidermal proteins SAX-7, MNR-1, and muscle secreted LECT-2) to the intracellular actin cytoskeleton (*Figure 8*, left). Previous studies showed that this process required the HPO-30 ICD to bind to the WRC, which in turn could activate Arp2/3 to generate branched actin networks to deform the membrane and promote new dendrite formation

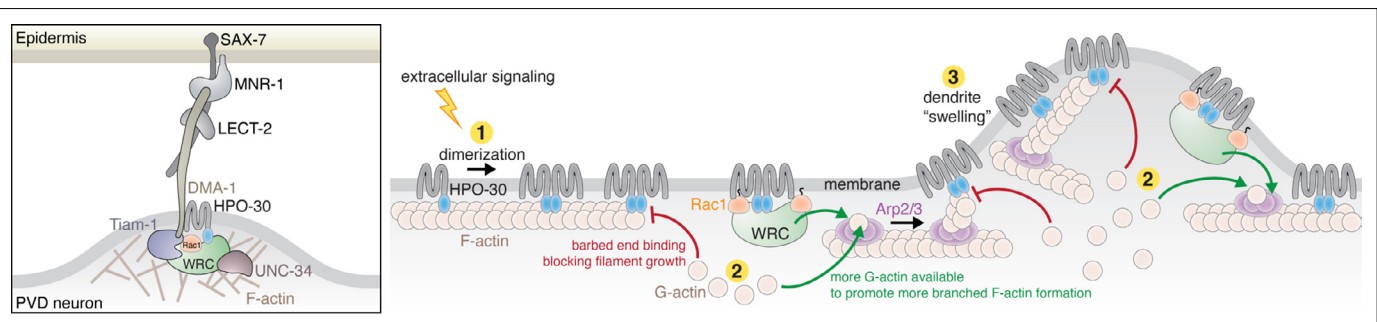

**Figure 8.** HPO-30 intracellular domain (ICD) integrates two distinct mechanisms to regulate actin dynamics during dendrite branching. Left: In *C. elegans* PVD neurons, a variety of molecules work together with the HPO-30/DMA-1 co-receptor to link extracellular signals to the intracellular actin network critical for driving dendrite branching. Right: Model of how HPO-30 ICD modulates actin dynamics. In (1), HPO-30 on the dendrite membrane undergoes dimerization (or oligomerization). Both monomeric and dimeric forms of HPO-30 ICD can bind to the side of actin filaments, but only the dimeric ICD can bind to the fast-growing barbed end. In (2), dimeric ICD binding to the barbed end acts as a capping protein, which stops long filament growth and helps reserve G-actin for the Arp2/3 complex to produce short, branched actin filaments. In (2), dimeric ICD can also interact with the WAVE regulatory complex (WRC). This recruits the WRC to the membrane and simultaneously promotes WRC activation by Rac1, which in turn stimulates Arp2/3 to produce branched actin filaments. In (3), the dual actions of HPO-30 ICD facilitate the formation of branched actin networks, which can help generate the 'swelling' of dendrite observed in previous studies (*Shi et al., 2021*), an important indicator of the outgrowth of a new dendrite branch.

(*Zou et al., 2018*). In this study, our multifaceted analyses reveal two novel mechanisms employed by the HPO-30 ICD to regulate local actin dynamics. First, the ICD undergoes dimerization to bind the WRC, which, in addition to recruiting the WRC to local dendrite branching sites, can cooperate with Rac1 to enhance WRC activation (*Figure 8*, step 2). This mechanism can work synergistically with DMA-1, as the cytoplasmic tail of DMA-1 directly recruits the Rac-GEF, TIAM1, which would increase the local concentration of active GTP-Rac1 (*Zou et al., 2018*). Second, the HPO-30 ICD directly binds to actin filaments to modulate actin dynamics, with monomeric and dimeric forms binding to the side of filaments, and the dimeric ICD alone binding to the barbed end to inhibit both actin polymerization and depolymerization, similar to CapZ activity (*Figure 8*, steps 1 and 2). These seemingly contradictory actin regulatory mechanisms may enable exquisite control of local actin dynamics necessary for dendrite branching (*Figure 8*, right, and see discussion below).

HPO-30 presents a unique example of a transmembrane protein capable of interacting with the WRC, distinct from the previously identified WIRS peptide-containing receptors such as SYG-1, Robo, DCC, Neogenin, TMEM132, neuroligins, and various protocadherins (*Chaudhari et al., 2021*; *Chia et al., 2014*; *Fan et al., 2018*; *Lee et al., 2016*; *Wang et al., 2021*; *Xing et al., 2018*). Unlike these receptors, the HPO-30 ICD does not use a WIRS motif or a similar short peptide motif to bind WRC. Instead, our structure-function analyses show that the entire ICD sequence is necessary for binding to the WRC, and dimerization of the ICD is crucial for this interaction. This novel WRC-binding mechanism should be conserved across animals, as the HPO-30 ICD is present only in nematodes but can bind to both *C. elegans* and human WRCs (*Zou et al., 2018*). Thus, HPO-30 represents a new class of transmembrane proteins that interact with the WRC using a distinct mechanism from the WIRS-containing receptors identified previously (*Chen et al., 2014b*).

Without a high-resolution structure of the ICD complexed with WRC, the exact conformation that the ICD adopts and how dimerization mediates WRC binding remain to be determined. Our computational and biophysical analyses consistently indicate that the monomeric ICD contains transient helical structure, but it is overall predominantly disordered in solution. Two possible mechanisms could explain how the ICD binds to WRC and why dimerization appears important for the interaction. In the first mechanism, the ICD may have one binding surface on the WRC. The dimerized ICD acts as one structural entity, which undergoes induced folding and adopts a stable conformation to bind the WRC. Induced folding has been proposed for many DNA-binding proteins and cell-signaling molecules, such as the interaction between E-cadherin and β-catenin or between *Tg*IST and STAT1 (*Dyson and Wright, 2002*; *Huang et al., 2022*; *Huber et al., 2001a*; *Huber and Weis, 2001b*; *Turjanski et al., 2008*). In the second mechanism, the ICD may have two separate binding surfaces on the WRC. Dimerization tags attached to the ICD provide an avidity effect that promotes simultaneous binding of two copies of ICD to the two surfaces on the same WRC. Because ICD can directly bind to another ICD, we favor the first mechanism, but our data cannot rule out the second mechanism. The above two mechanisms can also explain how the ICD binds to the barbed end of actin filaments in a dimerization-dependent manner. Regardless of which mechanism is used, it is likely the ICD adopts two distinct conformations when binding to WRC vs. binding to the barbed end of F-actin. Such structural plasticity has been observed for signaling molecules with versatile functions (*Bürgi et al., 2016*; *Dishman and Volkman, 2018*; *Okuda et al., 2022*). Resolving the high-resolution structures and identifying the key residues responsible for each binding event would allow us to differentiate the two functions of HPO-30 ICD more precisely.

The requirement of ICD dimerization for both WRC-binding and F-actin capping suggests an intriguing mechanism of how HPO-30 regulates actin remodeling at dendrite branching sites. Clustering of membrane receptors is a common mechanism to increase local signaling molecule concentration (*Johannes et al., 2018*). The enrichment of HPO-30 at developing PVD neuron dendrites correlates with the high level of F-actin in these branches (*Zou et al., 2018*). In addition to increasing its local membrane density, clustering of HPO-30 may facilitate dimerization of its ICD, which can in turn activate its ability to recruit the WRC and cap actin filaments. This dimerization-mediated functional switch can help the cell to distinguish signal from noise and achieve sharp spatiotemporal control of actin polymerization in response to upstream stimuli. It is worth noting that the regions of the ICD that most dramatically affect its ability to interact with another ICD are the same regions that are most important for binding to WRC and binding/capping F-actin. This again support the idea that dimerization (or oligomerization) is key to ICD function.

Our study indicates that the HPO-30 ICD has two functions upon binding to the WRC: (1) recruiting the WRC to dendrite branching site membranes and (2) enhancing WRC activation by Rac1. This dual action of binding and sensitizing the WRC has been observed in several WIRS-containing receptors, such as PCDH10 and PCDH19, where WIRS peptide binding recruits the receptor to the membrane while weak interactions from the flanking ICD sequences promote WRC activation by Rac1 (*Chen et al., 2014b*). Since the HPO-30 ICD strongly inhibits actin polymerization, it will be important to develop new assays to directly measure WRC activation, such as tracking WCA release, to understand how HPO-30 ICD binding enhances WRC activation by Rac1. As Rac1 activates the WRC by binding to two distinct sites located on the Sra1 subunit (*Chen et al., 2017*; *Ding et al., 2022*), it is possible that HPO-30 ICD promotes Rac1-WRC activation by directly stabilizing Rac1 binding to either site or by destabilizing WCA sequestration.

To our knowledge, HPO-30 represents the first transmembrane protein that possesses actin capping (or capping-like) activity. The precise structural mechanism of how the dimeric HPO-30 ICD binds to the barbed end of actin filaments remains to be determined. It is possible that it binds to the same surface as CapZ. Alternatively, it might bind to a nearby surface that could either sterically clash with or allosterically destabilize CapZ binding. Regardless, the dual abilities of HPO-30 to promote WRC membrane recruitment and activation while also capping actin filaments and inhibiting actin polymerization may seem contradictory at first glance. However, capping activity, which is usually provided by cytosolic capping proteins like CapZ, is known to be critical for Arp2/3-based actin dynamics in both bead motility assays in vitro and many processes in vivo (*Akin and Mullins, 2008*; *Miyoshi et al., 2006*). Capping proteins can stop long actin filament growth and simultaneously reserve G-actin by blocking its access to barbed ends (*Carlier and Pantaloni, 1997*). In parallel, capping protein can also prevent the barbed end from sequestering the WH2 sequence of WCA, which would otherwise deplete the activity of WASP family proteins (*Co et al., 2007*; *Higgs et al., 1999*). Together, by blocking the barbed end, the capping activity stimulates the formation of new branches through Arp2/3 and thus plays a crucial role in dynamically modulating the actin architecture to generate force against the surface where actin is localized (*Akin and Mullins, 2008*; *Funk et al., 2021*; *Figure 8*, step 2). In principle, the capping activity of the HPO-30 ICD should act similarly to stimulate the formation of a highly branched actin network. Although the isolated ICD has apparently lower capping activity compared to the canonical capping protein CapZ, it has the advantage of concentrating the capping activity in the area immediately next to membranes, which can provide a more focused control of actin dynamics.

In summary, our study establishes HPO-30 as a novel membrane receptor that, upon dimerization, can directly integrate the activities of WRC-Arp2/3-mediated actin polymerization and actin filament capping, distinguishing it from other WIRS-containing receptors and canonical barbed end cappers like CapZ. These two distinct functions of HPO-30 provide an intriguing model to explain how it regulates local actin dynamics to facilitate dendrite branching in PVD neurons (*Figure 8*). The synergistic action of both functions would promote the formation of highly branched actin networks, which help generate dendritic 'swellings' observed in PVD neurons immediately before dendrite branch outgrowth (*Figure 8*, step 3; *Shi et al., 2021*). This mechanism likely works in conjunction with other actin machinery in the cell to achieve highly spatiotemporally controlled actin dynamics and neuronal cell branching.

## Materials and methods
### Protein expression and purification

See *Supplementary file 1* for a.a. sequences of recombinant proteins used in this study. HPO-30 ICD proteins (and their associated alanine scan mutants), including GST-ICD, ICD-GST, DLC8-ICD-His6, GB1-ICD-His6, His9-sumo-ICD, MBP-ICD, GST-ICD-mEGFP, GB1-FKBP-ICD-His6, GB1-FRB-ICD-His6, GB1-FKBP-His6, GB1-FRB-His6, SNAP-FKBP-ICD-His6, SNAP-FRB-ICD-His6, SNAP-FKBP-His6, and SNAP-FRB-His6, as well as other related proteins, including MBP-WAVE1 (1-230), MBP-WAVE1 (1-230)-WCA, MBP-WAVE1 (1-178), MBP-Abi2 (1-158), MBP-HSPC300, (MBP)₂-Abi2 (1-158), (MBP)₂-HSPC300, and Rac1$^{Q61L/P29S}$ Δ4 (herein referred to as Rac1$^{QP}$), were individually expressed in Arctic Express (DE3) RIL (Agilent) or BL21 (DE3) T1$^R$ (Sigma) cells after induction with 0.5–1 mM IPTG at 10°C or 18°C for 16 hr. His10-SNAP-CapZ β and CapZ α were expressed together from the pCDFDuet

vector (Novagen) in Arctic Express (DE3) RIL (Agilent) cells after induction with 0.75 mM IPTG at 10°C for 16 hr. GST-ICD or ICD-GST (and alanine scan mutants) and GST-mEGFP were purified through Glutathione Sepharose beads (GE Healthcare), followed by cation exchange chromatography using a Source 15S column (GE Healthcare) at pH 7.0. The GST-ICD tryptophan mutations ('black' W22X) were purified through Glutathione Sepharose beads (GE Healthcare), followed by dialysis into pull-down buffer containing 100 mM NaCl. MBP-ICD (and associated W22R 'black' mutant) was purified through amylose resin (New England Biolabs), followed by cation exchange chromatography using a Source 15S column (GE Healthcare) at pH 7.0. The GST-tag from GST-ICD-mEGFP and the MBP tag from MBP-ICD were removed using TEV cleavage at 4°C overnight, followed by cation exchange chromatography using a Source 15S column (GE Healthcare) at pH 7.0. DLC8-ICD-His6, GB1-ICD-His6, His9-sumo-ICD, GB1-FKBP-ICD-His6 (and associated alanine mutants), GB1-FRB-ICD-His6 (and associated alanine mutants), GB1-FKBP-His6, GB1-FRB-His6, GB1-His6, GB1-homoCC-ICD-His6, SNAP-FKBP-ICD-His6, SNAP-FRB-ICD-His6, SNAP-FKBP-His6, and SNAP-FRB-His6 were purified through Ni-NTA Agarose resin (QIAGEN), followed by cation exchange chromatography using a Source 15S column (GE Healthcare) at pH 7.0. The SNAP-FKBP-ICD-His6 and SNAP-FRB-ICD-His6 alanine mutants were purified through Ni-NTA Agarose resin (QIAGEN), followed by dialysis into 50KMEH5Gd. GST-ICD, ICD-GST, DLC8-ICD-His6, GB1-ICD-His6, His6-sumo-ICD, ICD-mEGFP, untagged ICD, untagged ICD W22R, GB1-FKBP-ICD-His6 (and associated alanine mutants), GB1-FRB-ICD-His6 (and associated alanine mutants), GB1-FKBP-His6, GB1-FRB-His6, GB1-His6, GB1-homoCC-ICD-His6, SNAP-FKBP-ICD-His6, SNAP-FRB-ICD-His6, SNAP-FKBP-His6, SNAP-FRB-His6 were further purified through a Superdex 75 column (GE Healthcare). His10-SNAP-CapZ β and CapZ α were purified through Ni-NTA Agarose resin (QIAGEN), followed by anion exchange chromatography using a Source 15Q column (GE Healthcare) at pH 8.0 and size exclusion chromatography on a Superdex 200 column (GE Healthcare). Rac1$^{QP}$Δ4 was purified by an SP-Sepharose Fast Flow column (GE Healthcare) at pH 7.0 followed by size exclusion chromatography through a Superdex 75 column.

MBP-WAVE1, MBP-Abi2, MBP-HSPC300, (MBP)$_2$-Abi2, and (MBP)$_2$-HSPC300 proteins were purified through amylose beads (New England Biolabs). The Sra1/Nap1 dimer was obtained by co-expressing His6-Sra1 and untagged Nap1 in Tni insect cells (Expression Systems), followed by purification through Ni-NTA agarose resin (QIAGEN) and anion exchange chromatography using a Source 15Q column at pH 8.0. Pentameric WRC was assembled and purified following previously described protocols (*Chen et al., 2014a*, *Chen et al., 2010*). Briefly, individually purified WAVE1, Abi2, and HSPC300 subunits were mixed at equimolar ratio in the presence of 1% (wt/vol) NP40 and incubated on ice for 48 hr. The assembled trimer was then purified by anion exchange chromatography through a Source 15Q column at pH 8.0 and cation exchange chromatography by a Source 15S column at pH 6.0. Dimer and trimer were mixed at equimolar ratio and incubated on ice for 30 min. The assembled pentamer was purified on amylose beads (NEB), after which the MBP and His6 tags were cleaved using TEV protease incubation overnight. The pentamer was further purified using anion exchange chromatography through a Source 15Q column at pH 8.0 and size exclusion chromatography using a Superdex 200 column.

Actin was purified as previously described from rabbit muscle acetone powder from Pel-Freeze (*Spudich and Watt, 1971*). Actin was labeled by pyrene or Alexa Fluor 647 after polymerization at 4°C, using a 10-fold or 2-fold excess dye, respectively. Actin and pyrene-labeled actin used in actin polymerization and depolymerization assays were kept in continuous dialysis at 4°C, with buffer changes twice a week. Actin and Alexa Fluor 647-labeled actin used in smTIRF experiments were kept at 4°C away from light for up to 2 weeks.

## Generation of WRC-(MBP)$_2$

To create WRC:Abi2-(MBP)$_2$, the sortase ligation sequence, LPGTG, was genetically fused to the C-terminus of MBP-Abi2 (1-158). Meanwhile, a TEV site was added to the N-terminus of an (MBP)$_2$ tag, which after Tev cleavage would expose a Gly required for sortase ligation. MBP-Abi2 (1-158)-LPGTG was expressed, purified, and incorporated into the WRC as described above to create WRC-LPGTG. GG-2MBP was expressed in Arctic Express (DE3) RIL (Agilent) cells after induction with 0.75 mM IPTG at 10°C for 16 hr, purified on amylose resin, and subjected to TEV cleavage overnight, followed by anion exchange chromatography using a Source 15Q column (GE Healthcare). Sortase5M (sortase A pentamutant) was a gift from David Liu (Addgene plasmid # 75144), expressed in Arctic Express (DE3)

RIL (Agilent) cells, purified over Ni-NTA agarose resin, followed by cation exchange using a Source 15S column (GE Healthcare) and size exclusion chromatography using a Superdex 75 column (GE Healthcare) (*Chen et al., 2011*). WRC-LPGTG (1 µM) was mixed with GG-MBP (25 µM) and sortase (10 µM) in 50 mM Tris pH 7.5, 150 mM NaCl, and 10 mM CaCl$_2$ and left at RT for 2 hr. The reaction was quenched by adding 25 mM EGTA, and the WRC-(MBP)$_2$ was purified over a Superdex 200 column to separate the WRC-(MBP)$_2$ from unligated molecules and sortase.

## Regular pull-down assay

GST pull-down assays were performed as previously described (*Shi et al., 2021*). Briefly, 20 µL of GSH-Sepharose beads were mixed with bait protein and prey protein in 1 mL of pull-down buffer (50 mM NaCl, 10 mM HEPES pH 7.0, 5% (wt/vol) glycerol, 5 mM 2-mercaptoethanol, and 0.05% Triton X-100). For reactions at alternative pH values, either 10 mM MES pH 6.0 or 20 mM Tris pH 8.0 was used in place of HEPES. The samples were mixed at 4°C for 30 min, washed three times with 1 mL of pull-down buffer, and eluted with 40 µL of elution buffer containing 30 mM reduced glutathione and 100 mM Tris pH 8.5. MBP pull-down assays were performed like GST pull-down assays, except that 20 µL of amylose resin was used, and elution buffer included 2% (wt/vol) maltose. His-tagged pull-down assays were performed as above, used 20 µL of Ni-NTA agarose resin, G-Buffer (2 mM Tris-HCl pH 8.0, 200 µM ATP, 0.5 mM DTT, 0.1 mM CalCl$_2$, and 1 mM NaN$_3$) as the wash buffer. The elution buffer contained 500 mM imidazole pH 7.0. In all pull-down assays, the eluate was examined by SDS-PAGE and Coomassie blue staining. In all pull-down assays using FKBP and FRB, rapamycin was added to 5 µM final concentration in all buffers. In control reactions, the same volume of DMSO was added in place of rapamycin.

For the WRC alanine scan pull-down quantification (*Figure 1B*), the intensity of the Sra1 and Nap1 bands were quantified using ImageJ. The intensity from the GST control lane was subtracted from the alanine protein lane, and the corrected intensity was divided by the intensity of the GST-ICD band. This ratio was then divided by the ratio of the WT HPO-30 lane. For the MBP-ICD alanine scan pull-down quantification (*Figure 4A*), the intensity of the MBP-ICD band was quantified using ImageJ and divided by the intensity of the GST-ICD band. The ratio of the GST control was then subtracted from the ICD ratio. The corrected ratio was then set relative to the WT ICD lane. For both the WRC and MBP-ICD alanine scan pull-downs, ANOVA on anks and Dunn-Tukey tests were performed to determine significance.

## EPD assay

EPD assays were performed as previously described (*Chen et al., 2017*). Briefly, 60 µL of GSH-Sepharose beads (50% slurry equilibrated in pull-down buffer) were mixed with 0.1 µM WRC and various amounts of GST-tagged protein (from 0.01 µM to 30 µM) and brought to 100 µL final reaction volume using pull-down buffer (composition the same as in GST pull-down assays, above). The reactions were allowed to mix for 30 min at 4°C, and four reactions at a time were spun at 15 krpm for 15 s. The supernatant was removed and examined by SDS-PAGE and Coomassie blue staining. Each assay was repeated three times. The Sra1/Nap1 intensity was quantified using ImageJ to calculate the fractional occupancy. The data from all repeats were pooled and globally fitted in DynaFit using a single binding site model (*Kuzmic, 1996*).

## Size exclusion chromatography analysis

GB1-FKBP-ICD and GB1-FRB-ICD were mixed at equimolar ratio and loaded onto a 24 mL Superdex 200 column (GE healthcare) equilibrated in 100 mM NaCl, 10 mM HEPES pH 7.0, 5% (wt/vol) glycerol, and 1 mM DTT, with or without 5 µM rapamycin. Molecular weight was determined by running molecular weight standards (Sigma Cat # MWGF200-1KT) in identical buffer as the ICD proteins and generating a standard curve from the elution volumes.

## Pyrene-actin polymerization assay

Pyrene-actin polymerization assays were performed as previously described (*Doolittle et al., 2013a*). Actin was purified and pyrene-labeled as described above and kept in continuous dialysis in G-Buffer (2 mM Tris-HCl pH 8, 0.5 mM DTT, 0.2 mM ATP, 0.1 mM CaCl$_2$, 1 mM NaN$_3$) that was changed twice a week. Arp2/3 was purified following existing protocols and kept aliquoted at –80°C (*Doolittle et al.,*

*2013b*). All proteins except for the WRC and Arp2/3 were purified into 50KMEH5Gd (50 mM KCl, 1 mM MgCl₂, 1 mM EGTA, 10 mM HEPES pH 7.0, 5% [wt/vol] glycerol, 1 mM DTT) and stored at –80°C. WRC230VCA was purified into 100KMEI20Gd (100 mM KCl, 10 mM imidazole pH 7.0, 20% [wt/vol] glycerol) and kept at –80°C. Unless otherwise noted, a typical reaction contained 2 µM actin with 10% pyrene labeled, 10 nM Arp2/3, 100 nM WRC[230WCA] or free WCA, and/or 0.4 µM Rac1[QP]Δ4, and/or additional ICD ligands to be analyzed, with or without 5 µM rapamycin or an equivalent volume of DMSO. For reactions in *Figure 5C*, reactions contained 4 µM actin with 5% pyrene labeling, with 5 µM rapamycin. The excitation and emission wavelengths were set to 365 nm and 407 nm, respectively, and data was collected every 2 s for 30 min. Pyrene quenching assays (*Figure 5—figure supplement 1C*) were performed by mixing 4 µM actin with 5% pyrene labeling, either in the monomeric state or pre-polymerized by the addition of 50KMEH5Gd, with ICD and monitoring fluorescence over time. The long-term actin assays (*Figure 5—figure supplement 1E*) were assembled the same as for *Figure 5C*, but data was collected every 2 min over the course of 16 hr. The polymerization rate at $t_{1/2}$ was obtained by programs at https://biochempy.bb.iastate.edu/, which uses the same algorithm as described in *Padrick et al., 2008*. Data were collected on a TECAN SPARK plate reader.

## Actin depolymerization assay

Actin depolymerization assays were performed as previously described, with some modifications (*Heiss and Cooper, 1991*). Actin at ~20 µM and 70% pyrene labeling was pre-polymerized at RT overnight by addition of 1 mM MgCl₂ and 50 mM KCl. For each depolymerization reaction, actin was diluted to 5 µM in one pooled mixture, either alone or with proteins to be tested, and left for 3 min at RT to allow for protein binding. Protein at the same concentration was prepared in a second, separate mixture. After 3 min, the actin was further diluted 20-fold by the addition of the second mixture. All proteins were diluted into depolymerization buffer, which is three parts G-Buffer and one part 50KMEH5Gd (see above for buffer composition). The excitation and emission wavelengths were set to 365 nm and 407 nm, respectively. Data were collected on a TECAN SPARK plate reader. To calculate the relative initial velocity, the slope for the first 30 s of the reaction was calculated and divided by the slope of the actin control. ANOVA on Ranks and Dunn-Tukey tests were performed to determine significance.

## CD measurement

Untagged HPO-30 ICD was purified into 100 mM NaCl, 10 mM HEPES pH 7.0, 5% (wt/vol) glycerol, and 1 mM DTT. The same buffer was used to dissolve BSA powder (Fisher Cat # BP1600-100) and to blank the CD spectrometer. The blank buffer was degassed by sonication for 3 min at 30% power. Data were collected on a MOS-500 CD spectrometer using an ALX250 lamp. Data were collected for wavelengths between 190 nm and 260 nm, with a 1 nm step, 0.5 s acquisition period, and averaged over three repeats. Mean residue ellipticity was calculated as described previously (*Greenfield, 2006*). Data for BSA and HPO-30 were collected at the same concentration of protein. TMAO was dissolved to 3 M in the same buffer as HPO-30 and was added to the appropriate concentration before measurement. Separate buffer blanks containing the same concentration of TMAO were measured to ensure TMAO alone did not contribute to the spectrum.

## NMR spectroscopy

Isotopically labeled proteins were expressed and purified as described for non-labeled proteins, using minimal media containing either N[15] NH₄Cl or N[15] NH₄Cl and C[13] glucose instead of traditional media. Proteins were purified into 100 mM NaCl, 10 mM HEPES pH 7.0, 5% (wt/vol) glycerol, 1 mM DTT and were supplemented with 10% D₂O. Protein concentration ranged from 70 µM (GB1 tag) to 850 µM (GB1-ICD). GB1 tag was produced by thrombin cleavage of GB1-ICD and removal of ICD using nickel-NTA resin (QIAGEN). NMR spectra were collected on a Bruker 700 MHz spectrometer at Iowa State University equipped with z-shielded gradient triple resonance 5 mm TCI cryoprobe. 2D ¹H-¹⁵N TROSY-HSQC and ¹H-¹⁵N HSQC experiments were recorded with a time domain matrix consisting of 100* ($t_1$, ¹⁵N)×1024* ($t_2$, ¹H) complex points with acquisition time of 50 ms ($t_1$) and 91.8 ms ($t_2$) using 16 scans per FID and 1.5 s interscan delay. Spectral widths for ¹H and ¹⁵N dimensions were set to 15.9 ppm and 28.2 ppm, respectively, with carriers set at 4.821 ppm (¹H) and 119.138 ppm (¹⁵N). Sequential ¹H/¹⁵N/¹³C backbone assignments of the ICD were achieved using conventional 3D triple resonance

correlation experiments (HNCO, HNCA, and HN(CO)CA) at 700 MHz. Secondary $^{13}$Cα chemical shifts were calculated using Poulsen's random coil database with corrections for pH, temperature, and neighbor a.a. correction (*Kjaergaard et al., 2011a*; *Kjaergaard and Poulsen, 2011b*). All spectra were processed using NMRPipe (*Delaglio et al., 1995*) and displayed with SPARKY (*Lee et al., 2015*).

## Fluorophore labeling of proteins

SNAP-tagged proteins were labeled with SNAP-Surface Alexa Fluor 488, SNAP-Surface Alexa Fluor 546, and SNAP-Surface Alexa Fluor 647 (New England Biolabs). Protein (5 µM) and dye (10 µM) were mixed and allowed to react in 50KMEH5Gd at RT for 2 hr. Labeled proteins were desalted into 50KMEH5Gd buffer and concentrated. Extinction coefficients of fluorophores were calculated from a standard curve and are as follows: Alexa 488 at 495 nm, 95,000 M$^{-1}$*cm$^{-1}$; Alexa 546 at 556 nm, 120,000 M$^{-1}$*cm$^{-1}$; Alexa 647 at 650 nm, 255,000 M$^{-1}$*cm$^{-1}$. Protein labeling efficiency was calculated by dividing protein concentration by dye concentration—for Alexa Fluor 488 the labeling efficiency was estimated at ~100%, for Alexa Fluor 546 the labeling efficiency was estimated at ~60%.

## smTIRF data collection

All time lapses were collected on a Nikon TE2000-E inverted microscope equipped with a 100× 1.49 NA TIRF objective and a TIRF Quad filter cube (Chroma C141789), using an Andor iXon3 EM-CCD (DU-897-CS0) camera, with a GATACA iLas system to prevent uneven illumination. Coverslips were prepared as described previously with slight modifications (*Narvaez-Ortiz and Nolen, 2022*). Briefly, glass coverslips (VWR Cat # 48393-241) were cleaned with 2% (wt/vol) Hellmanex, acetone, and 1 M KOH solutions with sonication, and rinsed extensively with DI water before each step and after the KOH treatment. Coverslips were rinsed with methanol and dried using an N$_2$ gas stream. GOPTES (Fisher Cat # G0210100G) was added to the coverslips, which were then baked at 75°C for 30 min. Coverslips were rinsed with microscope-grade acetone and dried with N$_2$ gas stream. A 2% (wt/wt) Biotin-PEG3000/NH2-PEG3000 mixture of powder (Rapp Polymere GmbH) was prepared, placed on top of the coverslips, and the coverslips baked overnight at 75°C. After overnight baking, the coverslips were washed with water and dried with N$_2$ gas. Individual wells were made by placing functionalized coverslips on Ibidi sticky-Slide IV 0.4 slides (Ibidi Cat # 80608). Individual lanes were prepared for microscopy by incubating with 5% (wt/vol) Pluronic F-127 for 10 min at RT, followed by either 100 nM (for elongation rate analysis) or 35 nM (for side binding analysis) neutravidin incubation (in 50 mM Tris pH 7.5, 200 mM NaCl, referred to as LS TBS for short) for 10 min at RT, and either 10 nM (for elongation rate analysis) or 3.5 nM (for side binding analysis) biotinylated myosin (in LS TBS) incubation for 10 min at RT. The chambers were washed with 20 mg/mL BSA in 50 mM Tris pH 7.5, 600 mM NaCl, and incubated with 20 mg/mL BSA in LS TBS for 10 min at RT. Actin was diluted to a final concentration of 2.1 µM with 20% Alexa 647 labeling in 1× TIRF buffer (final concentrations: 50 mM KCl, 2 mM MgCl$_2$, 2 mM EGTA pH 8.0, 10 mM imidazole pH 7.0, 25 mM glucose, 1 mM Trolox, 0.5% methylcellulose [400 cP], 20 mM 2-mercaptoethanol, 0.4 mM ATP, 2 mg/mL BSA, 0.02 mg/mL catalase, 0.1 mg/mL glucose oxidase, 1 mM 4-nitrobenzyl alcohol, and 0.5 mM propyl gallate) and allowed to polymerize on the slide for 5 min at RT. Excess actin was removed by two washes, each time using 40 µL of 1× TIRF buffer. A separate mixture of 1 µM actin with 20% Alexa 647 labeling containing desired proteins in 1× TIRF buffer was then added to the wells to start data acquisition. Time lapse images were acquired using the following setups. Experiments involving unlabeled HPO-30 ICD and CapZ for elongation rate analysis of actin$^{AF647}$: 640 nm laser, 5% power, 50 ms exposure time, and a 5 s interval between exposures; experiments involving CapZ$^{AF488}$: 488 nm laser (15% power, 50 ms exposure time) and 640 nm laser (5% power, 50 ms exposure time), alternating between 20 consecutive exposures in the 488 nm channel for CapZ$^{AF488}$ and one exposure in the 640 nm channel for actin$^{AF647}$, with a 500 ms interval between exposures; experiments involving HPO-30 ICD$^{AF546}$: 561 nm laser (10% power, 50 ms or 100 ms exposure time) and 640 nm laser (5% power, 50 ms exposure time), alternating between 20 consecutive exposures in the 561 nm channel for ICD$^{AF546}$ and one exposure in the 640 nm channel for actin$^{AF647}$, with a 500 ms interval between exposures.

## smTIRF data processing—actin elongation rate measurement

Time lapses were opened in ImageJ and the background removed using a rolling ball radius of 10 pixels. The length of actin filaments was calculated using the Filament Length ImageJ plugin, kindly

provided by Jeff Kuhn, using a Gaussian radius of 1.5–2, determined for each individual movie (*Kuhn and Pollard, 2005*). The length (in µm) was converted to subunits using the established value of 370 actin subunits/µm (*Huxley and Brown, 1967*). Time points were taken from the NIS Elements software. Filaments were only selected if they were present within the first 10 frames of the movie and did not leave the frame during the course of the video. 10 time points were selected and the length at each point was calculated, and the average slope was used for the actin elongation rate. ANOVA with Dunn-Tukey tests were used to determine significance.

## smTIRF data processing—side binding/capping analysis

Time lapses were opened in ImageJ and the background removed using a rolling ball radius of 10 pixels. Only filaments present at the beginning of the videos and those that did not leave the frame during the duration of the video were selected. Analysis was performed in a single-blinded manner. Side binding events were determined if they met the following criteria: (1) the ICD/empty tag puncta must be present for more than one frame; (2) the ICD/empty tag puncta must move with the filament at least once; (3) the filament must not move away from the ICD/empty tag puncta; (4) the HPO-30/vector puncta must be smaller than a circle with a radius of 4 pixels. Capping events were confirmed by the absence of growth with puncta present and, if available, growth from the end capped after the puncta leaves the filament. A Student's t-test was used to determine significance between ICD and tag reactions.

## Actin pelleting assay

Actin pelleting assays were performed based on previous protocols (*Heier et al., 2017*) with slight modifications. Actin was pre-polymerized at room temperature (RT) overnight by addition of 1X50KMEH5Gd. Reactions (60 µL) were assembled by mixing 2 µM actin and 1 µM protein (in the same 50KMEH5Gd buffer), which were then allowed to bind at RT for 30 min. Reactions were centrifuged at $100,000 \times g$ at 4°C for 30 min in a Type 42.2 Ti rotor in a Beckman ultracentrifuge. 40 µL of the supernatant was removed and mixed with SDS-PAGE loading buffer, and the remaining ~15 µL was removed and discarded. The pellet was dissolved by the addition of 40 µL of G-buffer, followed by brief pipetting and vortexing, and allowed to sit at RT for 5 min before the liquid was removed and mixed with SDS-PAGE loading buffer. The intensity of the supernatant and pellet bands on SDS-PAGE gels were measured using ImageJ. The total intensity of the supernatant and pellet bands, and the percentage of intensity from the pellet and the supernatant were calculated. The percentage of pelleted protein was calculated by subtracting the percentage of intensity from the pelleted protein in the absence of actin from the percentage of intensity from the pelleted protein in the presence of actin. For example, in *Figure 6A*, the intensity of BSA from lanes 1 and 2 was summed and the percentage of intensity from lanes 1 and from 2 were calculated. This was repeated for lanes 3 and 4, then the percentage intensity of lane 2 was subtracted from lane 4. ANOVA with Dunn-Tukey tests were used to determine significance.

To quantify the dissociation constant of F-actin binding, 0.2 µM F/F-ICD was mixed with increasing concentrations of polymerized actin and pelleted as described above. The concentration of F-actin was estimated to be the same as the concentration of G-actin. 40 µL of the supernatant was removed and mixed with SDS for visualization on a gel. The intensity of both F/F-ICD bands was quantified using ImageJ to calculate the fractional occupancy. The data was fitted in DynaFit using a single binding site model (*Kuzmic, 1996*).

## Actin pelleting competition assays

Actin pelleting competition assays were performed nearly identically to the actin pelleting assay described above, except that after 30 min of incubation of 5 µM HPO-30 ICD proteins with 2 µM actin, 600 nM CapZ was added and incubated for 5 min before ultracentrifugation. The intensity of the top bands for CapZ in the pellet were calculated for all reactions and corrected for the intensity of CapZ pelleted without actin. The relative intensity was calculated by dividing the intensity of the lane by the intensity of the lane containing CapZ with actin alone. For example, in *Figure 7D*, the intensity of lane 2 was subtracted from lanes 4, 6, 8, and 10. Then, the corrected intensity of lanes 6, 8, and 10 was divided by the corrected intensity of lane 4. ANOVA with Dunn-Tukey tests were used to determine significance.

## Tryptophan fluorescence assays

Untagged ICD and GB1 tag were purified into 50 mM NaCl, 10 mM HEPES pH 7.0, 5% (wt/vol) glycerol, 5 mM 2-mercaptoethanol (called 50 PDB, for pull-down buffer). Tryptophan powder (Sigma Cat # 73-22-3) and BSA (Fisher Cat # BP1600-100) were dissolved in the same buffer. Solutions were made at the correct concentrations (ranging from 0 to 50 µM) and an emission scan was measured from 320 nm to 400 nm. The absorption wavelength was set to 243 nm to minimize the fluorescence value and prevent the inner-filter effect from non-specifically reducing the emission values (*Weitner et al., 2022*). The excitation and emission slit widths were set to 3 nm. All measurements were performed on a Horiba Scientific FluoroMax spectrofluorometer in 3 mm quartz cuvettes (Starna Cells, Inc Cat # 3-3.45-Q-3). Normalized peak fluorescent values were calculated by dividing the fluorescence value of the peak by the concentration of that sample.

Untagged ICD 'black' W22R was purified into 50 PDB. To measure the concentration, samples of a known concentration of WT ICD were run on a gel and used to make a standard curve. Experiments were performed as described above for ICD WT. To calculate the fluorescence of ICD WT+ICD 'black' W22R, the fluorescence of ICD 'black' W22R alone was subtracted from the fluorescence of ICD 'black' W22*R*+ICD WT. Extra NaCl was added to the 200 PDB samples to bring the final salt concentration to 200 mM NaCl.

## MD simulations

All-atom MD simulations were performed for each structural model using the Condo cluster at the High-Performance Computing Facility at Iowa State University. These simulations were conducted with GROMACS (*Abraham et al., 2015*) using the AMBER99SP-disp force field (*Robustelli et al., 2018*). The system was solvated in a triclinic simulation box with TIP4P water molecules. Bonds to hydrogen were constrained via the LINCS algorithm (*Hess et al., 1997*). Short-range electrostatic and Lennard-Jones interactions were calculated with a plain coulomb cutoff of 1.0 nm. The particle mesh Ewald scheme with grid spacing of 0.16 nm was utilized for long-range electrostatic interactions (*Essmann et al., 1995*). Solvent and solute were separately coupled to a modified Berendsen thermostat (velocity rescale) with a reference temperature of 300 K and Parinello-Rahman barostat with a reference pressure of 1 bar. Minimization and equilibration were performed for 1 ns coupled to a Berendsen thermostat (velocity rescale) with a reference temperature of 300 K and a reference pressure of 1 bar coupled to. After energy minimization and a 1 ns equilibration, the MD simulations were carried out over 200 ns with 2 fs timesteps and leap-frog integrator.

## S2 cell culture and transfection

*Drosophila* S2 cells were grown in Schneider's media (Thermo Fisher) supplemented with 10% heat-inactivated FBS (Life Technologies) and 50 U/mL penicillin-streptomycin (Thermo Fisher). Cells were transfected with Effectene (QIAGEN) and 1 µg of total plasmid (either Pactin>HPO-30:6xMyc (pXD384) or both Pactin>HPO-30:6xMyc (pXD384) and Pactin>HPO-30:HA (pXD226)).

## Co-immunoprecipitation

S2 cells were harvested 72 hr after transfection. Cells were lysed in RIPA buffer (Thermo Fisher) with 1× Halt Protease Inhibitor Cocktail (Thermo Fisher) for 30 min on ice. Cell lysates were incubated with anti-HA affinity gel beads (Sigma E6779-1ML) for 1 hr at 4°C with rotation. Proteins were eluted at 80°C in NuPAGE LDS Sample Buffer (Life Technologies) supplemented with DTT (GoldBio) and detected using western blot with mouse antibody to HA (1:1000, Sigma H3663), rabbit antibody to Myc (1:1000, Santa Cruz Biotechnology sc-789), and HRP-conjugated goat antibodies to mouse (1:20,000, Jackson Immuno Research).

## Acknowledgements

We thank ResearchIT at Iowa State University for installing AlphaFold 2 and AlphaFold Multimer; Daniel Boesch for help with submitting AlphaFold prediction tasks; Scott Nelson at Iowa State for guidance with CD spectroscopy; Xuefeng Wang at Iowa State for the initial setup of smTIRF experiments; Aubrey Sijo-Gonzales, Finlan Rhodes, Leyuan Loh, Ganesh Prasad, Simanta Mitra, and

ResearchIT at Iowa State University for establishing a web-based application (https://biochempy.
bb.iastate.edu) for data visualization; and Michael Rosen at UT Southwestern for sharing FKBP/FRB
constructs.

## Additional information

### Competing interests

Kang Shen: Reviewing editor, *eLife*. The other authors declare that no competing interests exist.

### Funding

| Funder | Grant reference number | Author |
|---|---|---|
| National Institutes of Health | Department of Biology | Baoyu Chen |
| Roy J. Carver Charitable Trust | | Baoyu Chen |
| National Institutes of Health | R35 GM136319 | Brad J Nolen |
| National Institutes of Health | R01 GM132561 | Julien Roche |
| Howard Hughes Medical Institute and the National Institute of Neurological Disorders and Stroke | R01 NS082208 | Kang Shen |
| National Science Foundation | Graduate Research Fellowship | Rebecca Shi |
| National Institutes of Health | | Rebecca Shi |

The funders had no role in study design, data collection and interpretation, or the decision to submit the work for publication.

### Author contributions

Daniel A Kramer, Conceptualization, Data curation, Formal analysis, Investigation, Visualization, Writing - original draft, Writing - review and editing; Heidy Y Narvaez-Ortiz, Data curation, Methodology, Writing - review and editing; Urval Patel, Rebecca Shi, Data curation, Investigation, Visualization, Writing - review and editing; Kang Shen, Resources, Supervision, Funding acquisition, Writing - review and editing; Brad J Nolen, Resources, Supervision, Funding acquisition, Investigation, Methodology, Writing - review and editing; Julien Roche, Conceptualization, Resources, Data curation, Formal analysis, Supervision, Funding acquisition, Investigation, Visualization, Methodology, Writing - review and editing; Baoyu Chen, Conceptualization, Resources, Formal analysis, Supervision, Funding acquisition, Investigation, Visualization, Methodology, Writing - original draft, Project administration, Writing - review and editing

### Author ORCIDs

Daniel A Kramer ⓘ http://orcid.org/0000-0002-7159-9562
Kang Shen ⓘ http://orcid.org/0000-0003-4059-8249
Brad J Nolen ⓘ http://orcid.org/0000-0002-0224-9980
Julien Roche ⓘ http://orcid.org/0000-0003-1254-1173
Baoyu Chen ⓘ http://orcid.org/0000-0002-6366-159X

### Decision letter and Author response

Decision letter https://doi.org/10.7554/eLife.88492.sa1
Author response https://doi.org/10.7554/eLife.88492.sa2

## Additional files

### Supplementary files
- Supplementary file 1. Amino acid sequences of recombinant proteins used in this study.
- MDAR checklist

### Data availability
All data generated or analyzed during this study are included in the manuscript and supporting files; Source Data files have been provided for Figures 1 to 7.

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
