## [Editor Report]

This important work investigates a dual mechanism of regulation of actin assembly by the dendrite branching receptor HPO-30. Convincing data demonstrate how oligomerization of HPO-30 ICD impacts the activity of the Wave Regulatory Complex (WRC), but also that direct binding of HPO-30 to filamentous actin modulates actin polymerization properties. This work sheds light on a crucial biological question but also shows the importance of the issue of conformation and oligomerization of disordered proteins in the regulation of actin regulatory mechanisms.

---

## [Decision Letter]

**[Editors’ note: the authors submitted for reconsideration following the decision after peer review. What follows is the decision letter after the first round of review.]**

**Decision letter after the first round of peer review:**

Thank you for submitting the paper "Dendrite branching receptor HPO-30 uses two novel mechanisms to regulate actin cytoskeletal remodeling" for consideration by *eLife*. Your article has been reviewed by 3 peer reviewers, one of whom is a member of our Board of Reviewing Editors, and the evaluation has been overseen by a Senior Editor. The following individuals involved in the review of your submission have agreed to reveal their identities: Pirta Hotulainen (Reviewer #2) and Yansong Miao (Reviewer #3).

Comments to the Authors:

We are sorry to say that, after consultation with the reviewers, we have decided that this work will not be considered further for publication by *eLife*.

Specifically, the Reviewers agreed on the fact that the dimerization of the ICD domain is not demonstrated. They acknowledge that the interaction of such disordered domains is complicated to study, but also that the current data are not sufficient to appreciate the detail of these intramolecular interactions. Furthermore, the effect of the ICD domain on actin polymerization is not entirely clear at this stage and would deserve additional experimental effort.

*Reviewer #1 (Recommendations for the authors):*

This manuscript investigates how the intracellular fragment (ICD) of the transmembrane protein HPO-30, involved in dendrite morphogenesis, interacts with the WAVE complex and regulates actin assembly.

The first part of the work shows that the ICD domain is itself unstructured and mostly inactive, but that its fusion to dimeric domains such as Gst tags increases its binding affinity to the WAVE regulatory subunits Sra1 and Nap1. Structural analysis by NMR and prediction software indicate that ICD is possibly composed of one α-helix and two β sheets which could dimerize into a structured domain in some conditions.

The second part of the work investigates how ICD domains promote WRC activation by the GTPase Rac1. The authors identify a dual effect of ICD, promoting both the activation of the Arp2/3 complex by WRC and Rac1 while binding to the side of actin filaments and inhibiting actin polymerization quite significantly.

This work is certainly important to the specialized community. The experiments are carefully conducted and well interpreted. However, a few important experiments are still missing to clearly demonstrate the dimerization of the ICD domain of HPO30 and to better understand its effect on actin assembly. This last comment is not meant to undermine the general quality of the work but suggests some further efforts to bring it to completion.

This work has potential but is still too preliminary at this stage:

1. Despite the authors' best efforts, the ability of ICD to dimerize is not formally demonstrated in this manuscript. The summary of the article shows that the authors are aware of the problem ('likely undergoes dimerization'), but the rest of the article is not so careful in its wording and misleads the readers. For a journal of the level of *eLife*, I think the authors should give a more precise answer to this problem.

2. I find the pull-down experiment in Figure 3F not very convincing. The authors must provide a control showing that the co-immunoprecipitation of HA and Myc-tagged proteins does not come from HPO being a transmembrane protein in this experiment.

3. About the pyrene actin experiments:

– Do ICD domains quench the fluorescence of pyrene when they bind to the side of actin filaments, as cofilin does?

– When the authors write that a given factor inhibits actin polymerization, they should specify whether they talk about rates of actin assembly or final amounts of polymer. To do this, they also need to provide pyrene curves on longer time scales and up to plateau values to assess steady-state levels of actin polymerization. Moreover, actin seems to polymerize really slowly in some experiments (for example, the black curve with 4 µM actin in Figure 4B). Why is that?

– For more clarity/simplicity, the authors should also provide data showing the effect of ICD constructs on actin polymerization only.

4. The authors must provide representative TIRF movies used for the quantification of Figure 6C and more details about how the movies are quantified. With a protein like CapZ, for example, we usually detect transitions between normal elongation and pauses where elongation is completely stopped. So the authors should be able to analyze two populations of filaments, one elongating normally and the other not. So do the measured speeds (sometimes up to 1 subunit per second) represent here averages or filaments continuously elongating at intermediate rates? Is something similar (normal elongation and pauses) observed with ICD or do filaments elongate at intermediate rates?

*Reviewer #2 (Recommendations for the authors):*

This study evaluates two new mechanisms of how dendrite branching receptor HPO-30 regulates actin polymerization. While it directly binds filamentous actin, both on side of the filament and to the polymerizing barbed end, it also activates WRC, thus promoting Arp2/3-complex induced actin branching and polymerization. Altogether, combining these two mechanisms, it is expected that HPO-30 promotes the formation of a highly branched network with a short filament length.

The strength of this study is that it uses numerous biochemical and structural methods to clarify how the HPO-30 receptor regulates actin cytoskeleton. Methods and results seem solid and they support each other. As such this is a nice piece of work and helps future studies to clarify what exactly happens in neurons and in dendrite branching.

I would be careful with interpretations of neurons without any neuronal data. Arp2/3 complex-induced actin polymerization is for sure an important part of dendrite branching but I personally don't believe that it is enough. Therefore, model figure 7 can be slightly misleading or at least incomplete.

*Reviewer #3 (Recommendations for the authors):*

Daniel Kramer and colleagues have described a work that particularly focuses on the conformation and dimerization (or oligomerization) of HPO-30 ICD, and the ICD-mediated inhibition of Rac1-activated WRC for actin polymerization. It is an excellent effort with many high-quality results that address a very important biological question: how the conformation and oligomerization of disordered proteins affect actin or actin regulatory proteins, thereby regulating actin polymerization through tuning the protein conformation or oligomerization. It could be a report with high scientific significance if the mechanisms behind it could be dissected and interpreted better. In general, the biochemical experiments were done with high quality. I acknowledge the technical challenges behind such aspects of studying this ICD and many other disordered proteins on how they remodel the protein interactions and biochemical activities due to the nature of the protein being dynamic in conformational change or molecular interactions. While I very much appreciate the research done in this work, I also feel the several experimental results end up with incoherent interpretation and overstatement and are unable to support their conclusions. Several results are misinterpreted with speculation or overstatement that could be misleading.

1. The way the author used to conclude (e.g.e line 165) could be misleading. Neither the discrepancy between different prediction algorithms nor the different conformations from an individual algorithm should be considered solid evidence to conclude the dynamic conformation of a protein. Experimental validation using a structural biology approach, such as the NMR the authors did, or at least an MD simulation that measures the probability of each conformation, is reasonable. Here, with the NMR results, may the authors elaborate on if these results support the conformational flexibility and secondary structures of ICD in each examined fusion? The authors did not mention the NMR results of GST-ICD in the results though it was shown in Figure 2D.

2. May the authors describe if ICD forms any secondary structure when a dimeric GST is attached? One in silico prediction-based conclusion, "HPO-30 ICD may use a folded structure, instead of a short peptide motif, to bind the WRC" was not supported by another result about ICD conformation without tags, HPO-30 ICD, which is disordered in solution. It leaves a major weakness of the manuscript, which is the mechanism of the "dimer" concept that is used by switching among ICD-tag fusion, ICD only, and tag only in the different experiments. By that, the author often uses the dimerized tag as a presumption to conclude a dimerization of ICD, which is misleading. Please see below.

For example, the GST introduced stability of the ICD (as data not shown) imply a stable conformation or conformational constrain that prevents the proteolytic enzyme function during the protein purification procedures. Hence, firstly, the mechanisms for binding between ICD could be because of the conformational changes of ICD that are introduced by GST-dimer/other-dimeric tag instead of dimerizing ICD. I appreciate the use of different dimeric sets to support the importance of having a dimer tag. Nevertheless, there is a logical issue behind the conclusion. (a) Dimerization of ICD refers to the intermolecular interaction between two ICD monomers by having physical contact sites. (b) The dimerization of GST only provides a tag to bring two ICD monomers together, which does not provide a dimeric interface between ICDs. (c) with the same logic, a dimeric form of any unstructured/structured proteins will not introduce inter-molecular interaction by just being a dimer if the monomer does not provide a binding.

Second, the authors did not show experimental evidence to prove that GST-ICD is a dimer. Dimerization is a presumption based on GST, the final oligomerization status and potential heterogeneity of oligomerization of GST-ICD need to be defined experimentally. Otherwise, the authors can only claim that the dimeric motif (GST or FKBP-FRB) can facilitate the binding between ICD and WRC. Considering the IDR nature with elongated shape, several approaches are able to define its oligomerization status and heterogeneity, including SEC-MALS, AUC, and fluorescent-labeling and imaging-aided single particle-imaging by using a set of proteins at specified oligomerization status to generate a standard curve.

Third, in vivo dimerization conclusion is an overstatement. Co-IP experiments only can support the interactions between bait and prey, which do not verify dimerization, oligomerization status, or heterogeneity. Hence, the conclusion supporting the hypothesis on Page 20, lines 422-425, is invalid.

1. For Figure 3 supplementary 1D, the authors need to provide a standard and calibration to indicate the calculated MW and oligomerization status for samples before and after adding rapamycin. In addition, do F and F-ICD fully overlap before adding rapamycin on Superdex 75?

2. The different interpretations of why F/F-ICD inhibits Rac1-activated WRC in actin polymerization appear confusing. On one side, the authors hypothesize that a direct binding between ICD and actin is the key (page 19, line 396). By the way, no inhibition of actin spontaneous polymerization by ICD was observed. On the other side, authors claim that the inhibitory effect of ICD in the reaction pool with mixed G-actin, F-actin, WRC, ICD, and Rac1, is an inhibition on the Rac1-activated WRC by citing an opposite activation result where a mixture of PCDH10 ICD activate actin polymerization in the presence of WRC and Rac1. I had difficulty understanding how these different messages support each other and how these conclusions were deduced from these related but conflicting information. I guess that F/F-ICD might have different interactions towards Rac1 or have a higher affinity towards the Rac1-activated WRC after undergoing the conformational changes, such as from either disordered- or ordered changes to another ordered status or changes in complex oligomerization status.

3. Direct binding of ICD to F-actin is an appealing phenomenon in F-actin sedimentation assay. For Figure 5, TIRF imaging would help to understand the binding or bundling at this high concentration of ICD with spatial information between ICD and F-actins. I also suspect that F/F-ICD likely bundles F-actin at such concentration. Rap induced more ICD in pellet at molar ratio ICD: G-actin >1, which has a much higher affinity to F-actin shown in 5 nM concentration TIRF assay in figure 6. These results suggest that either a high oligomerization of ICD happens on F-actin at this condition or F/F-ICD forms much-higher oligomer/aggregates in the solution without associating with F-actin.

Considering the impact of *eLife* in guiding future research, the current interpretation format would not be appropriate for future studies on the function and mechanism of disordered proteins.

Understanding ICD inhibition on the WRC-Rac1 complex would provide important knowledge. But the current result is not sufficient to understand the mechanism. Several key components are missing. 1. Quantitative biochemical characterization of inter and intramolecular interactions, such as Kd and oligomerization status, for single components or complex in the different combinations of ICD, WRC, Ras1, G-actin, or F-actin. If ICD is disordered, it will likely have weak interaction with binding partners, which could be technically challenging. Such weak interaction is reflected by the TIRF imaging, where the localization of ICD on F-actin is unstable, either sliding or with a rapid on/off rate. In addition, a higher-order oligomer (brighter dots) is not on the F-actin. I fully acknowledge the technical challenges behind such experiments to make a clear conclusion about the interactions involving the disordered region. I recommend examining the potential secondary structure behind such binding towards actin (G- or F-actin). Otherwise, a flexible interaction between a protein and an actin filament is hard to understand. One of the ways that might help us to understand it could be a surface charge. ICD is a positively charged polymer with a theoretical π of 9.18, which might provide non-specific interactions.

**[Editors’ note: the authors submitted for reconsideration following the decision after peer review. What follows is the decision letter after the second round of review.]**

**Decision letter after the second round of peer review:**

Congratulations, we are pleased to inform you that your article, "The intrinsically disordered cytoplasmic tail of a dendrite branching receptor uses two distinct mechanisms to regulate the actin cytoskeleton", has been accepted for publication in eLife.

Reviewer #1 (Recommendations for the authors): The authors have followed my recommendations from the previous round of review. I am in favour of publishing this study as is.

Reviewer #2 (Recommendations for the authors): I had only minor comments for the original manuscript and based on this comment authors have now revised the model Figure.

Reviewer #3 (Recommendations for the authors): In the revised manuscript by Daniel Kramer and colleagues, the focus is on understanding how HPO-30 ICD regulates WRC-mediated actin polymerization. The researchers aim to shed light on the role of ICD dimerization in inhibiting WRC-mediated actin assembly and its direct binding and regulation of the Rac1-WRC-actin complex. Previously, there were concerns about inconsistent interpretations and overstated results, which have now been addressed in the updated manuscript.

The strengths of this study include high-quality biochemical experiments and exploring a crucial biological question: the interaction of disordered proteins' conformation and oligomerization with actin and actin regulatory proteins, thus controlling actin polymerization. The revised manuscript tackles previous concerns and offers better insights into the mechanisms of ICD dimerization (or potentially oligomerization) and its influence on actin polymerization. This research adds valuable knowledge to the field by examining disordered proteins' effects on actin polymerization regulation and lays a solid foundation for future studies.

The authors have made significant efforts to substantiate the function derived from the dimerized ICD through multiple experiments, though they still cannot exclude the functional consequences from dynamic ensembles with varied oligomerization states. Nevertheless, this reflects the inherent nature of IDRs, which often evolve their assembly by gaining new interactions and altering their catalytic activities. This study effectively demonstrates the primary function of the major ensemble population, which frequently represents the dominant functional form in the crowded intracellular environment. Elucidating the fine-tuning functions associated with dynamic assemblies remains technically challenging.

Significant improvements have been made in the revised manuscript, offering a more coherent interpretation of results, clarifying IDR-mediated actin polymerization confusion, and addressing previous concerns.

This work holds the potential for a substantial impact on the field, as it investigates the function and mechanism of disordered proteins, a topic garnering increased interest in the scientific community. The methods and data presented in this study will be beneficial for researchers working on similar topics, serving as a basis for further exploration of disordered proteins' roles in regulating cellular processes.

The review comments were provided in the previous version, and the authors have adequately addressed my concerns. The revised version was improved significantly and deserved to be published by elife.

---

## [Author Response]

**[Editors’ note: The authors appealed the original decision. What follows is the authors’ response to the first round of review when sending the revised manuscript as a new submission for peer review.]**

Comments to the Authors:We are sorry to say that, after consultation with the reviewers, we have decided that this work will not be considered further for publication by eLife.Specifically, the Reviewers agreed on the fact that the dimerization of the ICD domain is not demonstrated. They acknowledge that the interaction of such disordered domains is complicated to study, but also that the current data are not sufficient to appreciate the detail of these intramolecular interactions. Furthermore, the effect of the ICD domain on actin polymerization is not entirely clear at this stage and would deserve additional experimental effort.Reviewer #1 (Recommendations for the authors):This manuscript investigates how the intracellular fragment (ICD) of the transmembrane protein HPO-30, involved in dendrite morphogenesis, interacts with the WAVE complex and regulates actin assembly.The first part of the work shows that the ICD domain is itself unstructured and mostly inactive, but that its fusion to dimeric domains such as Gst tags increases its binding affinity to the WAVE regulatory subunits Sra1 and Nap1. Structural analysis by NMR and prediction software indicate that ICD is possibly composed of one α-helix and two β sheets which could dimerize into a structured domain in some conditions.The second part of the work investigates how ICD domains promote WRC activation by the GTPase Rac1. The authors identify a dual effect of ICD, promoting both the activation of the Arp2/3 complex by WRC and Rac1 while binding to the side of actin filaments and inhibiting actin polymerization quite significantly.This work is certainly important to the specialized community. The experiments are carefully conducted and well interpreted.

We thank the reviewer for the supportive and insightful comments of the significance and quality of our work.

However, a few important experiments are still missing to clearly demonstrate the dimerization of the ICD domain of HPO30 and to better understand its effect on actin assembly. This last comment is not meant to undermine the general quality of the work but suggests some further efforts to bring it to completion.This work has potential but is still too preliminary at this stage:

We also appreciate the reviewer’s constructive critiques and thoughtful suggestions, which have helped us to strengthen the manuscript. As described in detail below, we have completed nearly all of the suggested experiments, as well as several additional experiments, and have made significant revisions to the manuscript accordingly. We are confident that our work is now complete and ready for publication.

1. Despite the authors' best efforts, the ability of ICD to dimerize is not formally demonstrated in this manuscript. The summary of the article shows that the authors are aware of the problem ('likely undergoes dimerization'), but the rest of the article is not so careful in its wording and misleads the readers. For a journal of the level of eLife, I think the authors should give a more precise answer to this problem.

We appreciate the reviewer's feedback on the ambiguity in our previous writing about the ICD’s ability to dimerize. To clarify this point, we have revised our writing and have included new data from several different approaches in the revised manuscript to support the ICD-ICD interaction. Please also see our response to Reviewer #3 for further details on this topic.

We have added a new section titled “HPO-30 ICD can self-associate” to the Results (Line 402) to emphasize the significance of the ICD-ICD interaction.

We have moved the previous pull-down figure in a supplementary figure (the old Figure 3— figure supplement 2C) to a main figure (new Figure 4A), which shows that purified GST-ICD can pull down MBP-ICD, suggesting that the ICD can directly interact with itself through dimerization (or even oligomerization).

We have included new data in the new Figure 4B,C, showing the influence of various alanine-scan mutants on the ICD-ICD interaction. Interestingly, we found that the same regions critical for WRC interaction are similarly critical for the ICD-ICD interaction.

Additionally, in the new Figure 4—figure supplement 4A, we have included new data showing the sensitivity of the ICD-ICD interaction to disruption by the pH of the buffer, which suggests the interaction is likely mediated by electrostatic interactions.

We made extensive and careful attempts to measure the binding affinity of the ICD-ICD interaction using the intrinsic fluorescence of tryptophan. We had to use low concentrations of protein and a sub-optimal wavelength to prevent non-specific decreases in fluorescence due to the inner filter effect. Despite this, our controls using the monomeric GB1 protein showed a similar increase in fluorescence with concentration, making it impossible to conclude if the same increase in the ICD is due to dimerization or some other, non-specific, concentration effect. We then explored alternative methods by screening for a functional, non-fluorescent ICD (in which all Trp, Tyr, and Phe were removed) to titrate into the WT ICD. Unfortunately, our control experiments show that the fluorescence changes observed could be attributed to non-specific background signals, as we saw the same increase in high salt buffer, which we know disrupts the interaction between ICD molecules. We have summarized these negative results in the new Figure 4—figure supplement 2 to report the technical difficulties of quantifying the binding affinity. Due to the small size and unstructured nature of the molecule and the low affinity between ICD-ICD, we were unable to try other methods suggested by Reviewer #3. Despite these limitations, we believe our qualitative data and mutagenesis data shown in Figure 4 and Figure 4—figure supplement 1 have provided adequate support that the ICD can self-associate.

2. I find the pull-down experiment in Figure 3F not very convincing. The authors must provide a control showing that the co-immunoprecipitation of HA and Myc-tagged proteins does not come from HPO being a transmembrane protein in this experiment.

We apologize for the potential misinterpretation of the results presented in the previous Figure 3F caused by our writing. The purpose of these data is to show that the full-length (FL) HPO-30 has the potential to dimerize (or oligomerize), thus indicating dimerization of HPO-30 could happen in cells. This figure is not intended to suggest if dimerization of FL HPO-30 occurs through a direct interaction between HPO-30 or through other mediating molecules, nor does it indicate if dimerization occurs through the transmembrane domain of HPO-30 or through the ICD. To avoid possible overinterpretation of these data, we have moved this figure to a supplemental figure (new Figure 4—figure supplement 1B) and modified our writing accordingly. Please also see our response to Reviewer #3 for further details on this topic.

3. About the pyrene actin experiments:– Do ICD domains quench the fluorescence of pyrene when they bind to the side of actin filaments, as cofilin does?

We appreciate this question. To address this concern, we have conducted additional experiments to show that ICD binding does not quench pyrene fluorescence from either G-actin or F-actin. These new data are presented in the new Figure 5—figure supplement 1C.

– When the authors write that a given factor inhibits actin polymerization, they should specify whether they talk about rates of actin assembly or final amounts of polymer. To do this, they also need to provide pyrene curves on longer time scales and up to plateau values to assess steady-state levels of actin polymerization.

We thank the reviewer for suggesting this experiment. Per the reviewer’s suggestion, we extended the duration of the actin polymerization assay to a much longer time scale. Our results demonstrate that HPO-30 ICD significantly reduces the actin polymerization rate, which is consistent with our single-molecule data. We have included these new data in the new Figure 5—figure supplement 1E.

Moreover, actin seems to polymerize really slowly in some experiments (for example, the black curve with 4 µM actin in Figure 4B). Why is that?

The rate of actin polymerization can vary depending on the age of the actin, which is a commonly observed phenomenon. We keep our purified G-actin in continuous dialysis conditions for several months, with fresh buffer changes twice a week. The polymerization ability of actin generally decreases over time, making it essential to only compare results obtained on the same day. We do not directly compare experiments obtained on different days or using different batches of actin. Critical experiments are typically repeated on different days to ensure we arrive at the same conclusions.

– For more clarity/simplicity, the authors should also provide data showing the effect of ICD constructs on actin polymerization only.

We appreciate the reviewer’s insightful suggestion and have conducted the experiment as suggested. Our new data, as shown in the new Figure 5C and Figure 5—figure supplement 1E, clearly demonstrate that the ICD inhibits spontaneous polymerization of actin in the absence of any other effectors.

4. The authors must provide representative TIRF movies used for the quantification of Figure 6C and more details about how the movies are quantified.

We thank the reviewer for bringing this to our attention. We would like to clarify that we did provide representative TIRF movies used for the quantification of Figure 6C (now new Figure 7C) in Video 5, which shows videos from all conditions side by side. We also provided detailed procedures of how we quantified these movies in the Materials and methods section of our manuscript. We hope this clarification resolves any prior misunderstandings.

With a protein like CapZ, for example, we usually detect transitions between normal elongation and pauses where elongation is completely stopped. So the authors should be able to analyze two populations of filaments, one elongating normally and the other not. So do the measured speeds (sometimes up to 1 subunit per second) represent here averages or filaments continuously elongating at intermediate rates? Is something similar (normal elongation and pauses) observed with ICD or do filaments elongate at intermediate rates?

We thank the reviewer for bringing up this discussion. We have added a new figure (new Figure 7E and Figure 7—figure supplement 2) to provide more detailed analysis of the differences in actin elongation rates between HPO-30 ICD and CapZ. The new figures provide representative kymographs and elongation curves (up to 10 events) for actin alone, with ICD, and with CapZ. For CapZ, we did observe transitions between normal elongation and a complete stop for long periods of time. In contrast, for HPO-30 ICD, we generally observed a slower rate (lower, intermediated slope) with less frequent pausing. We attribute this difference to the high affinity and very slow off rate of CapZ, compared to the low affinity of ICD.

Reviewer #2 (Recommendations for the authors):This study evaluates two new mechanisms of how dendrite branching receptor HPO-30 regulates actin polymerization. While it directly binds filamentous actin, both on side of the filament and to the polymerizing barbed end, it also activates WRC, thus promoting Arp2/3-complex induced actin branching and polymerization. Altogether, combining these two mechanisms, it is expected that HPO-30 promotes the formation of a highly branched network with a short filament length.The strength of this study is that it uses numerous biochemical and structural methods to clarify how the HPO-30 receptor regulates actin cytoskeleton. Methods and results seem solid and they support each other. As such this is a nice piece of work and helps future studies to clarify what exactly happens in neurons and in dendrite branching.

We appreciate the positive evaluation of the significance and quality of our work.

I would be careful with interpretations of neurons without any neuronal data. Arp2/3 complex-induced actin polymerization is for sure an important part of dendrite branching but I personally don't believe that it is enough. Therefore, model figure 7 can be slightly misleading or at least incomplete.

We appreciate the insightful comment from the reviewer about the complexity of dendrite branching, which we agree involves precise coordination of many different molecules. In response, we have revised the cartoon in Figure 7 to provide a more comprehensive depiction of other molecules known to be involved in PVD neuron dendrite branching, rather than solely focusing on HPO-30-WRC-Arp2/3. We hope this improved figure, now Figure 8, will not only help readers better understand the significance of our findings related to the HPO-30 ICD, but also place our work in the broader context of dendrite branching regulation. We acknowledge that other pathways may act in coordination with this pathway.

Reviewer #3 (Recommendations for the authors):Daniel Kramer and colleagues have described a work that particularly focuses on the conformation and dimerization (or oligomerization) of HPO-30 ICD, and the ICD-mediated inhibition of Rac1-activated WRC for actin polymerization. It is an excellent effort with many high-quality results that address a very important biological question: how the conformation and oligomerization of disordered proteins affect actin or actin regulatory proteins, thereby regulating actin polymerization through tuning the protein conformation or oligomerization. It could be a report with high scientific significance if the mechanisms behind it could be dissected and interpreted better. In general, the biochemical experiments were done with high quality.

We appreciate the reviewer's acknowledgement of the significance and quality of our work.

I acknowledge the technical challenges behind such aspects of studying this ICD and many other disordered proteins on how they remodel the protein interactions and biochemical activities due to the nature of the protein being dynamic in conformational change or molecular interactions. While I very much appreciate the research done in this work, I also feel the several experimental results end up with incoherent interpretation and overstatement and are unable to support their conclusions. Several results are misinterpreted with speculation or overstatement that could be misleading.

We appreciate the constructive feedback from the reviewer on how to improve our work. We have taken the reviewer’s comments into consideration and have conducted a series of new experiments, including nearly all of the experiments suggested by the reviewer. Additionally, we have extensively revised our writing to address potential incoherence, overstatement, misinterpretation, or misunderstanding. We hope the reviewer finds that these revisions have significantly strengthened our work.

1. The way the author used to conclude (e.g.e line 165) could be misleading. Neither the discrepancy between different prediction algorithms nor the different conformations from an individual algorithm should be considered solid evidence to conclude the dynamic conformation of a protein. Experimental validation using a structural biology approach, such as the NMR the authors did, or at least an MD simulation that measures the probability of each conformation, is reasonable. Here, with the NMR results, may the authors elaborate on if these results support the conformational flexibility and secondary structures of ICD in each examined fusion? The authors did not mention the NMR results of GST-ICD in the results though it was shown in Figure 2D.

We appreciate the reviewer’s insightful comments about the limitations of predicted structures and the importance of experimental evidence to support the structural dynamics of the ICD. To address these concerns, we have investigated the structural features of ICD by NMR through a near complete backbone chemical shift assignment as well as by extensive all-atom MD simulations, as suggested by the reviewer, to further evaluate the structural flexibility and dynamics of the ICD.

For the new NMR data, summarized in the new Figure 2D and Figure 2—figure supplement 1F,G, we recorded a set of triple resonance experiments (HNCA, HN(CA)CO, and HNCO) to assign the ^13^C backbone chemical shifts of the ICD and evaluate the degree of transient secondary structures. Our results showed little deviation (<1 ppm) from idealized random coil chemical shifts, indicating that the ICD is predominantly disordered in solution, which is consistent with our CD data and 2D NMR spectra. We did observe a small, but consistent positive secondary Cα chemical shift, suggesting the presence of a transient (~25-30%) alphahelical motif in the same region that was predicted to form an α helix in many of the *ab initio* predictions.

Furthermore, our new all-atom MD simulation results, summarized in the new Figure 1—figure supplement 3 and Figure 4—figure supplement 3C, demonstrate that HPO-30 ICD does not stably exist in any conformation predicted by various programs. Rather, it exists as a dynamic ensemble of various conformations throughout the simulations.

Together, these new experimental results corroborate the notion that the HPO-30 ICD alone is highly flexible and dynamic in solution.

2. May the authors describe if ICD forms any secondary structure when a dimeric GST is attached?

We thank the reviewer for bringing this to our attention and apologize for the confusion and misunderstanding caused. We did mention the NMR result shown in Figure 2D in Page 17, Line 352-359 of the previous manuscript. Upon re-examination, we realized that we described this result in a later part of the manuscript, which may have given the impression that we did not mention this result. We have revised our writing to mention this result upfront (Page 14, Line 250-254) and moved this result to the new Figure 2—figure supplement 1E to make it clearer. Our results showed that neither the monomeric GB1 tag nor the dimeric GST tag promoted ICD folding or secondary structure formation.

One in silico prediction-based conclusion, "HPO-30 ICD may use a folded structure, instead of a short peptide motif, to bind the WRC" was not supported by another result about ICD conformation without tags, HPO-30 ICD, which is disordered in solution.

We appreciate the reviewer's feedback on our *in silico* prediction-based conclusion about HPO30 ICD potentially using a folded structure to bind the WRC. We agree that this speculation lacked experimental support, and in the revised manuscript, we have replaced this assertion with a proposal of two equally possible mechanisms for the ICD-WRC binding (Page 49, Line 804821). In the first mechanism, the ICD may have one binding surface on the WRC. In this scenario, the dimerized ICD may act as one structural entity, which undergoes induced folding and adopts a stable conformation to bind the WRC. In the second mechanism, the ICD may have two separate binding surfaces on the WRC. In this case, dimerization tags attached to the ICD can provide an avidity effect that promotes simultaneous binding of two copies of ICD to the two surfaces on the same WRC. Because our data suggest ICD can directly bind to another ICD, we currently favor the first mechanism, but we acknowledge that our data cannot rule out the second mechanism. Until we can solve the high-resolution structure of the ICD-WRC complex, the binding mechanism will remain an open question.

It leaves a major weakness of the manuscript, which is the mechanism of the "dimer" concept that is used by switching among ICD-tag fusion, ICD only, and tag only in the different experiments. By that, the author often uses the dimerized tag as a presumption to conclude a dimerization of ICD, which is misleading. Please see below.

We appreciate this insightful comment and concur with the reviewer that using a dimerization tag, regardless of identity of the dimerization tag, does not guarantee ICD dimer formation. To address this concern, in our revised manuscript, we proposed two mechanisms that could explain the dimerization tag's function: (1) promoting ICD dimerization, or (2) promoting an avidity effect if the ICD has two distinct binding surfaces on the WRC. We still favor the ICD dimerization mechanism, as our data presented in the new Figure 4 and Figure 4—figure supplement 1 demonstrate that ICD can directly interact with each other, suggesting that ICD may indeed dimerize (or even oligomerize) and therefore it is plausible that the dimerization tags promote ICD dimerization. However, we acknowledge that without a high-resolution structure of the ICD-WRC complex, the precise mechanism of action will remain an open question.

For example, the GST introduced stability of the ICD (as data not shown) imply a stable conformation or conformational constrain that prevents the proteolytic enzyme function during the protein purification procedures.

We thank the reviewer for bringing this to our attention and apologize for any misunderstanding caused by our previous writing. We would like to emphasize that the GST-tag did not uniquely improve the stability HPO-30 ICD during protein purification. We have used various other affinity or solubility tags throughout the paper, which all produced ICD of similar quality. The only exception was when we observed obvious proteolysis in GST-tagged ∆8, ∆9, and ∆10 mutants (but not the wild type or other mutants), which led us to move the GST tag to the Cterminus of ICD to obtain full-length proteins for these mutants. Therefore, we do not think the GST tag uniquely stabilized the ICD. Moreover, the NMR spectrum of the ICD when attached to GST did not show any noticeable differences from the untagged ICD alone, as shown in the new Figure 2—figure supplement 1E. This suggests that the GST tag could not induce a conformational change in the ICD.

Hence, firstly, the mechanisms for binding between ICD could be because of the conformational changes of ICD that are introduced by GST-dimer/other-dimeric tag instead of dimerizing ICD. I appreciate the use of different dimeric sets to support the importance of having a dimer tag. Nevertheless, there is a logical issue behind the conclusion. (a) Dimerization of ICD refers to the intermolecular interaction between two ICD monomers by having physical contact sites. (b) The dimerization of GST only provides a tag to bring two ICD monomers together, which does not provide a dimeric interface between ICDs. (c) with the same logic, a dimeric form of any unstructured/structured proteins will not introduce inter-molecular interaction by just being a dimer if the monomer does not provide a binding.

We again appreciate the insightful comments/discussions regarding the distinction between dimeric tags and ICD dimerization. We hope that our responses to the preceding questions have provided ample clarification on this matter. Accordingly, we have revised our writing to reflect this distinction and explain why we favor the ICD dimerization model. Our data in the new Figure 4 and Figure 4—figure supplement 1 demonstrate that ICD self-association does occur, which supports the model that dimerization tags promote physical dimerization of ICD. However, we acknowledge that without a high-resolution structure of the ICD-WRC complex, we cannot precisely distinguish this model from others.

Second, the authors did not show experimental evidence to prove that GST-ICD is a dimer. Dimerization is a presumption based on GST, the final oligomerization status and potential heterogeneity of oligomerization of GST-ICD need to be defined experimentally. Otherwise, the authors can only claim that the dimeric motif (GST or FKBP-FRB) can facilitate the binding between ICD and WRC. Considering the IDR nature with elongated shape, several approaches are able to define its oligomerization status and heterogeneity, including SEC-MALS, AUC, and fluorescent-labeling and imaging-aided single particle-imaging by using a set of proteins at specified oligomerization status to generate a standard curve.

We appreciate the helpful suggestions from the reviewer on experimentally determining the dimerization (or oligomerization) state of the ICD-ICD interaction. Pease see our detailed responses to Reviewer #1 for more information on this topic. We have provided evidence in the new Figure 4 and Figure 4—figure supplement 1A supporting direct ICD-ICD interaction, which is likely mediated by electrostatic interactions and is mediated by the same sequences important for WRC binding. While we considered the experiments suggested by the reviewer, due to the small size and disordered nature of the ICD as well as instrumental limitations, we ultimately decided to determine the binding affinity by measuring the intrinsic fluorescence from the single tryptophane residue in the ICD. Unfortunately, despite pursuing this approach for nearly three months, we were unable to obtain a definitive answer due to various experimental complications, which we explained in our responses to Reviewer #1 and summarized in the new Figure 4—figure supplement 2. Nevertheless, despite the lack of quantitative measurements of the affinity, we are confident that our new data have provided sufficient evidence supporting the direct interaction between ICD molecules.

Third, in vivo dimerization conclusion is an overstatement. Co-IP experiments only can support the interactions between bait and prey, which do not verify dimerization, oligomerization status, or heterogeneity. Hence, the conclusion supporting the hypothesis on Page 20, lines 422-425, is invalid.

We agree with the reviewer's comment that co-IP experiments cannot confirm the dimerization, oligomerization, or heterogeneity of a protein complex. We apologize as our writing gave the impression that we seemed to rely on the co-IP experiment to verify the ICD dimerization. We want to clarify that we used co-IP as an auxiliary test to investigate whether HPO-30 has the potential to self-associate in cells, as we could not purify the full-length HPO-30. We did not arrive at the conclusion of ICD dimerization solely based on the co-IP experiment. Instead, we provide experimental evidence supporting direct ICD-ICD interaction using purified proteins, as shown in the new Figure 4 and Figure 4—figure supplement 1A. Please also see our response to Reviewer #1 for further details on this topic.

1. For Figure 3 supplementary 1D, the authors need to provide a standard and calibration to indicate the calculated MW and oligomerization status for samples before and after adding rapamycin.

We appreciate this suggestion. We have added standard proteins and calculated Mw, which are now shown in Figure 3—figure supplement 1C.

In addition, do F and F-ICD fully overlap before adding rapamycin on Superdex 75?

Yes, we can confirm that the two separate monomeric parts of F/F-ICD (i.e., GB1-FKBP-ICD and GB1-FRB-ICD) have very similar Mw and fully overlap before they were mixed with rapamycin in gel filtration. We have included gel filtration chromatograms obtained from largescale purification of the individual proteins in the new Figure 3—figure supplement 1E as supporting evidence.

2. The different interpretations of why F/F-ICD inhibits Rac1-activated WRC in actin polymerization appear confusing. On one side, the authors hypothesize that a direct binding between ICD and actin is the key (page 19, line 396). By the way, no inhibition of actin spontaneous polymerization by ICD was observed.

We appreciate the insightful question from the reviewer. Please see our responses to Reviewer #1 for information on the same topic. In response to this question, we have now provided new data in Figure 5C and Figure 5—figure supplement 1E, showing that the ICD inhibits the spontaneous polymerization of actin in the absence of any effector proteins.

On the other side, authors claim that the inhibitory effect of ICD in the reaction pool with mixed G-actin, F-actin, WRC, ICD, and Rac1, is an inhibition on the Rac1-activated WRC by citing an opposite activation result where a mixture of PCDH10 ICD activate actin polymerization in the presence of WRC and Rac1. I had difficulty understanding how these different messages support each other and how these conclusions were deduced from these related but conflicting information.

We apologize for any confusion caused by our previous writing and appreciate the reviewer for pointing this out. We have now revised our manuscript to provide clearer explanations regarding the distinct activities of HPO-30 ICD in the actin polymerization assay. Our actin polymerization data in the old Figure 4 (new Figure 5) support two points. First, ICD strongly inhibits actin polymerization independent of WRC, WCA, or Arp2/3. Second, ICD binding to WRC promotes WRC activation by Rac1 (Figure 5E). This activating effect is dependent on WRC, because ICD does not have any effect on the activity of isolated WCA in the same conditions (Figure 5F). However, the strong inhibitory effect of ICD on actin polymerization, particularly at high ICD concentrations, masks its activating effect on the WRC, thus complicating our data presentation.

Regarding the comparison to the WIRS-containing receptor PCDH10, we use PCDH10 ICD as an example to demonstrate that such cooperativity observed between ICD and Rac1 is not unprecedented. We had no intention to infer that PCDH10 ICD shares the same mechanism as HPO-30 ICD in promoting Rac1-WRC activity.

We hope our new writing and explanation have addressed the reviewer's concerns and provided clarity on this matter.

I guess that F/F-ICD might have different interactions towards Rac1 or have a higher affinity towards the Rac1-activated WRC after undergoing the conformational changes, such as from either disordered- or ordered changes to another ordered status or changes in complex oligomerization status.

We thank the reviewer for this insightful suggestion. We have performed an equilibrium pulldown experiment to test whether the ICD indeed prefers to bind the Rac1-activated WRC. Our new results in the new Figure 5 —figure supplement 1F demonstrate that the binding between ICD and the WRC was indeed enhanced by the presence of Rac1, as indicated by the reduced WRC signal in the supernatant of the pull-down reactions. These results suggest that the ICD preferentially interacts with the activated WRC, consistent with our observation that the ICD promotes WRC activation by Rac1.

3. Direct binding of ICD to F-actin is an appealing phenomenon in F-actin sedimentation assay. For Figure 5, TIRF imaging would help to understand the binding or bundling at this high concentration of ICD with spatial information between ICD and F-actins. I also suspect that F/F-ICD likely bundles F-actin at such concentration. Rap induced more ICD in pellet at molar ratio ICD: G-actin >1, which has a much higher affinity to F-actin shown in 5 nM concentration TIRF assay in figure 6. These results suggest that either a high oligomerization of ICD happens on F-actin at this condition or F/F-ICD forms much-higher oligomer/aggregates in the solution without associating with F-actin.

We appreciate the reviewer for the insightful analysis and pointing out the possibility of actin bundling activity. In our smTIRF imaging shown in Figure 7 and Video 3-5, we only observed events of the ICD binding to the side (or end) of F-actin, but not any clear bundling of F-actin, even when we increased the concentration of ICD from 15 nM (fluorophore-labeled) to 1 µM (unlabeled). To make a fair comparison between the single molecule experiment and the bulksolution assay, we have repeated the F-actin sedimentation assay using 1 µM ICD (new Figure 6A). At this lower concentration, the ICD still effectively co-pelleted with F-actin. Combining these observations, we conclude that while the ICD directly binds to the side or end of actin filaments, it does not bundle actin under our experimental conditions.

Considering the impact of eLife in guiding future research, the current interpretation format would not be appropriate for future studies on the function and mechanism of disordered proteins.

We appreciate the reviewer for recognizing the potential impact of our study on the field of disordered proteins. We have taken all the critiques raised by the reviewers seriously and addressed them point-by-point. We have also completed almost all the experiments suggested by the reviewers, as well as several other new experiments, which together have significantly strengthened our work. Moreover, our extensively revised writing now provides a more accurate presentation of our findings. Based on these improvements, we hope that our revised manuscript has met the high standards required for publication in *eLife*.

Understanding ICD inhibition on the WRC-Rac1 complex would provide important knowledge.

We apologize again for any misunderstanding our previous writing may have caused. Our study supports two distinct mechanisms of the ICD: (1) inhibition of actin polymerization, and (2) enhancement of WRC-Rac1 activation.

But the current result is not sufficient to understand the mechanism. Several key components are missing. 1. Quantitative biochemical characterization of inter and intramolecular interactions, such as Kd and oligomerization status, for single components or complex in the different combinations of ICD, WRC, Ras1, G-actin, or F-actin.

We appreciate the reviewer's comment on the importance of quantitative measurements of molecular interactions for a full understanding of the function of ICD on the WRC and actin. In response, we have included quantification of the binding affinity between the ICD and WRC (new Figure 1D) and between the ICD and F-actin (new Figure 6—figure supplement 1C). Although we attempted extensively to quantify the binding affinity of the ICD-ICD interaction using intrinsic tryptophan fluorescence measurements, we faced technical challenges and could not obtain definitive results. Nonetheless, our qualitative pull-down data and mutagenesis data in the new Figure 4 and Figure 4—figure supplement 1A support the notion that the ICD-ICD interaction is weak and sensitive to buffer conditions. We believe that these data are sufficient to support the two mechanisms proposed for how HPO-30 ICD regulates the actin cytoskeleton.

If ICD is disordered, it will likely have weak interaction with binding partners, which could be technically challenging. Such weak interaction is reflected by the TIRF imaging, where the localization of ICD on F-actin is unstable, either sliding or with a rapid on/off rate. In addition, a higher-order oligomer (brighter dots) is not on the F-actin. I fully acknowledge the technical challenges behind such experiments to make a clear conclusion about the interactions involving the disordered region. I recommend examining the potential secondary structure behind such binding towards actin (G- or F-actin). Otherwise, a flexible interaction between a protein and an actin filament is hard to understand. One of the ways that might help us to understand it could be a surface charge. ICD is a positively charged polymer with a theoretical π of 9.18, which might provide non-specific interactions.

We appreciate the insightful feedback and suggestions regarding the nature of ICD-actin interaction. We also acknowledge the technical challenges in studying interactions involving disordered regions, and we agree that the weak interaction between ICD and actin presents such a challenge. However, we believe that such interactions are not necessarily difficult to understand, as flexible or disordered sequences can often mediate important protein-protein interactions, as seen in many other systems. For example, the WIRS peptide-WRC interaction, WCA-Arp2/3 interaction, and most SH2 and SH3 binding peptides all involve flexible or disordered sequences. Disordered regions were also shown to mediate actin binding for actin binding proteins like juxtanodin and calponin.

We agree with the reviewer that the high positive charge of HPO-30 ICD might contribute to non-specific interactions. However, we believe that the interaction between ICD and actin is more specific in nature. First, the ICD only interacts with F-actin, but not with G-actin, suggesting a specific interaction surface only present on the polymerized actin. Furthermore, in the new Figure 6—figure supplement 1D,E, we have provided new data using alanine scan mutants and actin pelleting assay to identify the regions important for F-actin binding. Moreover, in the new Figure 6—figure supplement 1F,G, we have provided new data showing how the alanine scan mutants affect the capping activity using the F-actin depolymerization assay. These results show that the same regions that are important for WRC binding or ICD binding are also the most important for F-actin binding and the capping activity. Together, our data support the conclusion that the interaction between the ICD and F-actin is specific in nature.

In summary, we appreciate the thoughtful comments from the reviewer and have taken them into account in our revised manuscript. We believe that our new data support the importance of specific regions of the ICD in its interactions with actin and WRC, and we hope that our revised manuscript will provide a clearer understanding of the mechanism of ICD-mediated regulation of the actin cytoskeleton.